# Trusting the Untrustworthy: A Cautionary Tale on the Pitfalls of Training-based Rejection Option

## Abstract

We consider the problem of selective classification, also known as rejection option. We first analyze state-of-the-art methods that involve a training phase to produce a selective classifier capable of determining when it should abstain from making a decision. Although only some of these frameworks require changes to the basic architecture of the classifier, by adding a module for selection, all methods necessitate implementing modifications to the standard training procedure and loss function for classification. Crucially, we observe two types of limitations affecting these methods: on the one side, these methods exhibit poor performance in terms of selective risk and coverage over some classes, which are not necessarily the hardest to classify; and surprisingly, on the other side, the classes for which they attain low performance vary with the model initialization. Additionally, some of these methods also decrease the accuracy of the final classification. We discuss the limitations of each framework, demonstrating that these shortcomings occur for a wide range of models and datasets. Moreover, we establish a formal connection between the trade-off of detecting misclassification errors and the minimization of risks in selective classification. This connection enables the development of a testing framework that requires no training and can be seamlessly applied to any pre-trained classifier, thereby enabling a rejection option.

## 1 Introduction

In many applications, incorrect decisions can have severe consequences. Therefore, detecting and preventing them is crucial. Consequently, significant efforts are being made in various areas of artificial intelligence to enhance the reliability of automatic systems, as they are known to be prone to errors (e.g., in computer vision (Gao et al., 2022; Cobb & Looveren, 2022), in autonomous driving (Amodei et al., 2016; Bicer et al., 2020), in NLP (Jin et al., 2022; Carlini et al., 2021), and in medical analysis (Subbaswamy & Saria, 2020; Bernhardt et al., 2022)).

Avoiding wrong decisions by abstaining has been investigated in the field of artificial intelligence since its early stages (Chow, 1957; Flores, 1958; Chow, 1970; Pudil et al., 1992). Abstentions or rejections can be broadly categorized into two groups (Hendrickx et al., 2021): ambiguity, where the model is unable to replicate the optimal decision for certain inputs (Hellman, 1970; Fukunaga & Kessell, 1972), and novelty, where inputs at the test time are significantly different from those encountered during training (Vasconcelos et al., 1995; Seo et al., 2000; Vailaya & Jain, 2000). In this paper, we consider the problem of enhancing the reliability of a model by incorporating a *rejection option*. Standard models are designed to provide answers related to the task they have learned when presented with input samples. By incorporating a rejection option, these models have the ability to abstain from providing a decision when deemed too risky.

Clearly, abstention raises the question of the trade-off between reducing the risk of making wrong decisions while keeping the number of abstentions as low as desired, therefore maintaining data coverage. While current state-of-the-art train-based rejection option methods achieve great performance in controlling the global risk based on target coverage, a non-negligible imbalance is observed when looking at their performance class by class. Figure 1 shows the coverage and risk for each class for our method (Gini) and baseline training base methods. We observe that some classes have unreasonably low coverage and high risk, which is undesirable.

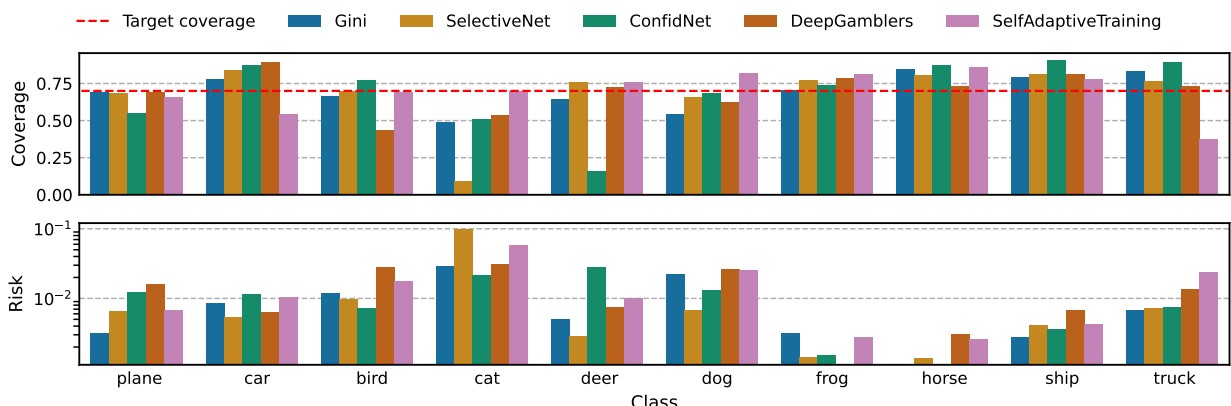

Figure 1: The performance of training-based methods often falls short, resulting in cases where the coverage (top) is low for at least one class, but the risk (bottom) is high. This undermines the reliability of these methods.

This work aims to shed a light on the causes of this unwanted behavior and to identify alternative principled approaches that mitigate disparities across classes.

**Main contributions.** Our contribution is threefold:

1. Our findings show two key aspects: train-based models for selective classification consistently perform well across the entire dataset, but they exhibit significant variations in terms of risk and coverage across different classes. This phenomenon is persistent on multiple runs of the same model with different initialization. We investigate and propose insights on the reason for this variation in risk and coverage across classes (see Section 3). Additionally, we note that some of the train-based methods also tend to decrease the performance in terms of classification accuracy (see Figure 2).

2. We make a mathematical connection between risk minimization in selective classification and error minimization in the context of misclassification detection. We prove that the optimal misclassification detector that achieves the best feasible trade-offs between type I and type II errors provides also the minimum selective classification risk (see Section 4). To the best of our knowledge, such a formulation has not been formalized in prior literature.

3. Finally, applying the result in the previous point, we implement and evaluate a rejection option method based on state-of-the-art post-hoc misclassification detector which can be applied to any pre-trained classifier (see Section 4.3), attaining favorable results.

We support these observations by presenting empirical results that involve several models (ResNet-34, DenseNet-121, and VGG-16) and benchmarks (CIFAR-10, CIFAR-100, and SVHN) (see Sections 5 and 6).

## 2 Related works

The growing body of work in deep learning has led to a renewed interest in the problem of decision rejection. Initially, researchers have been focusing on new metrics to assess the confidence of a model. Hendrycks & Gimpel (2017); Geifman & El-Yaniv (2017) propose to use the maximum of the softmax distribution output by a model as a confidence score on its decisions; Jiang et al. (2018) introduce a trust score which is proportional to the agreement between the considered model and a nearest-neighbor algorithm modified to only account for confident decisions; Gal & Ghahramani (2016) model the uncertainty using Monte Carlo Dropout to estimate the posterior predictive network distribution by sampling many stochastic network predictions.

More recent works in the field have focused on embedding the concept of rejection option directly at training time. These approaches propose either updated loss functions to account for the measure of risk related to accepting or rejecting a decision Liu et al. (2019) or a 'selection architecture' that returns an abstention output alongside the classification output Corbière et al. (2019), or both Geifman & El-Yaniv (2019). Huang et al. (2020) introduce a method to enhance the generalization of deep models through empirical risk minimization, specifically when dealing with corrupted data. By contributing to the calibration of the model's output during training, and adding a class to represent abstention, this technique is reported to better fit the correct data, increasing the chances of rejecting samples on which the confidence is on the low spectrum. This result is also supported by Fisch et al. (2022).

Rabanser et al. (2022) introduce a framework that, for a given test input, monitors the disagreement with the final predicted label over the intermediate models obtained during training. Although no active training is required, these frameworks need all the side information contained in the training dynamics. For both works, the code release is still pending. Granese et al. (2021) introduce a new simple state-of-the-art framework for misclassification detection which builds and improves on Hendrycks & Gimpel (2017). They apply a modified version of the Rényi Entropy to obtain a score for each input sample which is then used to decide whether to accept or reject the decision relative to the sample itself. This method does not require any training since it only uses the soft-probabilities output by the model, and will be formally linked to selective classification.

For the sake of completeness, we reference two recent studies related to the problem of selective classification, i.e., Schreuder & Chzhen (2021) and Gangrade et al. (2021b). The former focuses on fairness and adds the concept of demographic parity to the risk and coverage loss functions as an additional requirement. This results in a more balanced rejection rate for underrepresented groups in datasets such as Adult Income and German credit risk, where minority groups are present, at the cost of accuracy loss. As to the latter, to the best of our efforts due to the lack of official published code, we have not been able to reproduce the comparison with Geifman & El-Yaniv (2019); Liu et al. (2019) and add the comparison with Corbière et al. (2019) which is missing in Gangrade et al. (2021b). Feng et al. (2022) revisits Hendrycks & Gimpel (2017), and they have proposed to further regularize popular objective functions with entropy-minimization at training time. Finally, we mention important theoretical results Herbei & Wegkamp (2006); Franc et al. (2021); Fischer et al. (2016), and Cao et al. (2022) where the authors show the intrinsic equality between standard classification and selective classification by addition of a class representing abstention. Moreover, we observe how the interesting topic of rejection option crossed the boundaries of other well-established research areas such as certified robustness Cohen et al. (2019); Tramèr (2022), adversarial examples detection Aldahdooh et al. (2021), distribution shift Snoek et al. (2019), conformal prediction Einbinder et al. (2022), an extension of cost-sensitive selective classification (Liu et al. (2019)) to a larger family of loss functions Charoenphakdee et al. (2021), and selective classification in a more relaxed scenario, where extra side information in the shape of classification feedback is provided in case of abstention Gangrade et al. (2021a). Finally, we reference Zhang et al. (2023) as the most up to date survey paper on the topic.

## 3 Background on rejection option

Our study shows that training-based frameworks for rejection options often have inconsistent performance when evaluated on multiple instances of the same experiment. These methods may have adequate risk and coverage rates overall but may reject samples from specific classes excessively. Crucially, this behavior appears to be dependent upon changes in the model initialization. Although some classes are naturally harder to classify than others, the unwanted behavior mentioned above sheds light on the sensitivity of the frameworks to the initialization of the underlying model. We propose a brief analysis of popular train-based frameworks to understand these limitations.

Let us consider a standard classification task, where $\mathcal{X} \subseteq \mathbb{R}^d$ is the feature space and $\mathcal{Y} = \{1, \ldots, C\}$ is the label space. Let $\mathcal{D}_n = \{(\mathbf{x}_i, y_i)\}_{i=1}^n \sim p_{XY}$ denote the training set as a random realization of $n$ i.i.d. samples according to $p_{XY}$, the underlying and unknown probability density function over $\mathcal{X} \times \mathcal{Y}$. Let us define the *predictor* (i.e., the classifier) as $f_{\mathcal{D}_n}(\mathbf{x}) = \arg\max_{y \in \mathcal{Y}} P_{\widehat{Y}|X}(y|\mathbf{x}; \mathcal{D}_n)$ where $P_{\widehat{Y}|X}$ is the soft-prediction of the class posterior probability given a sample. For clarity, we define a soft-probability model that is associated with the predictor $f_{\mathcal{D}_n}$, s.t. $h_{\mathcal{D}_n} : \mathcal{X} \to \mathbb{R}^C$, s.t. $h_{\mathcal{D}_n,y}(\mathbf{x}) \in (0,1)$ and $\sum_{y=1}^C h_{\mathcal{D}_n,y}(\mathbf{x}) = 1$, i.e. $h_{\mathcal{D}_n}(\cdot)$

outputs $P_{\widehat{Y}|X}$[1]. The predictor is usually trained with the cross-entropy (CE) loss. Let $S : \mathcal{X} \to \{0,1\}$ be the *selector* which is responsible for rejecting/accepting the decision made by the *predictor*.

The selective model for a sample $\mathbf{x} \in \mathcal{X}$ is defined as:

$$(f_{\mathcal{D}_n}, S)(\mathbf{x}) \triangleq \begin{cases} f_{\mathcal{D}_n}(\mathbf{x}) & \text{if } S(\mathbf{x}) = 1 \\ \emptyset & \text{otherwise,} \end{cases} \tag{1}$$

where $\emptyset$ indicates that $f_{\mathcal{D}_n}$ abstains from the prediction.

We measure the performances of the selective model in terms of empirical coverage (Geifman & El-Yaniv, 2017; 2019) (the higher the better):

$$\hat{\phi}(S; \mathcal{D}_m) \triangleq \frac{1}{m} \sum_{i=1}^{m} S(\mathbf{x}_i), \tag{2}$$

and in terms of empirical selective risk (Geifman & El-Yaniv, 2017; 2019) (the lower the better):

$$\hat{r}(f_{\mathcal{D}_n}, S; \mathcal{D}_m) \triangleq \frac{\sum_{i=1}^{m} \mathbb{1}_{[f_{\mathcal{D}_n}(\mathbf{x}_i) \neq y_i]} S(\mathbf{x}_i)}{\sum_{i=1}^{m} S(\mathbf{x}_i)}, \tag{3}$$

where $\mathcal{D}_m = \{(\mathbf{x}_i, y_i)\}_{i=1}^{m}$ is the test or evaluation set and $\mathbb{1}_{[\cdot]}$ denotes the indicator function.

Instead of relying on the observation of the post-training performance of the model to decide whether a decision for an input sample should be accepted or rejected, train-based methods embed the rejection option within the training phase imposing the use of continuous and differentiable functions for the selection, combined with architecture restructuring Geifman & El-Yaniv (2019); Corbière et al. (2019) and convex combinations of multiple loss functions, or upgraded versions of the CE loss function that take into account a $|C| + 1$-th class that corresponds to the rejection option Liu et al. (2019); Huang et al. (2020). Our main observation is that, while these methods appear promising from a theoretical perspective, in practice they require significant tuning to achieve near-optimal solutions on heterogeneous datasets. This tuning requires additional samples to optimize the hyper-parameters involved in the training process. Without this consideration, the performance of these methods may exhibit unpredictable and undesirable behavior.

**SelectiveNet** (SN): Geifman & El-Yaniv (2019) propose to train their selective classifier by minimizing the loss function $\mathcal{L}_{SN}(h, S_{SN}, h', c; \mathcal{D}_n) = \alpha \times A + (1 - \alpha) \times B$, where $A = r(h, S_{SN}; \mathcal{D}_n) + \lambda \times \Psi(S_{SN}, c; \mathcal{D}_n)$, $B = \frac{1}{m} \sum_{i=1}^{m} \ell(h'(\mathbf{x}_i), y_i)$, $S_{SN} : \mathcal{X} \to [0,1]$ represents a soft selector, $\phi(S_{SN}; \mathcal{D}_n) \triangleq \frac{1}{n} \sum_{i=1}^{n} S_{SN}(\mathbf{x}_i)$, $r(h, S_{SN}; \mathcal{D}_n) \triangleq \frac{\frac{1}{n} \sum_{i=1}^{n} \ell(h(\mathbf{x}_i), y_i) S_{SN}(\mathbf{x}_i)}{\phi(S_{SN}; \mathcal{D}_n)}$ is the soft risk, $\Psi(S_{SN}, c; \mathcal{D}_n) = \max\{0, (c - \phi(S_{SN}; \mathcal{D}_n))^2\}$ denotes the constraint for the target coverage $c$, $\ell : Y \times Y \to \mathbb{R}^+$ is the loss function related to the classification task (e.g., cross-entropy), and $h' : \mathcal{X} \to \mathbb{R}^C$ is implemented by an auxiliary model that shares the same body of the model implementing $h$ but has an independent prediction head. Finally, $\lambda$ and $\alpha$, are fixed to 32 and 0.5, respectively. Our conjecture is that the optimization of the loss function $\mathcal{L}_{SN}$ given a specific amounts of training epochs, and changing the model's initialization causes the gradient descent to reach different local minima. This issue is exacerbated by the lack of validation for the parameters $\lambda$ and $\alpha$. In particular, the case presented in Figure 1 shows how SelectiveNet exhibits a much more conservative coverage than the other methods for the class "cat" in CIFAR-10, without containing the risk. Figure 4 shows that this is not uncommon, as both coverage and risk have many outlier results when considering the same experiment with 10 different random initializations.

**ConfidNet** (CN): given a pre-trained predictor $f$, Corbière et al. (2019) minimize the loss $\mathcal{L}_{CN}(S_{CN}; h, \mathcal{D}_n) = \frac{1}{n} \sum_{i=1}^{n} (S_{CN}(\mathbf{x}_i) - h_{y_i}(\mathbf{x}_i))^2$, where, during training, $S_{CN} : \mathcal{X} \to [0,1]$ is the learned selector, and $h_{y_i}$ represents the confidence of $h$ for the prediction $f$ over the ground truth class. In a nutshell, the training algorithm learns to predict the confidence of the model by analyzing the true posterior class probability of supervised samples. According to the theoretical framework introduced in Corbière et al. (2019), inputs with

---

[1]To simplify the notation, in the remaining of the work we will use $f$ and $h$ interchangeably with $f_{\mathcal{D}_n}$ and $h_{\mathcal{D}_n}$, respectively.

a low true class probability (smaller than $\frac{1}{C}$) are misclassified, while inputs with a true class probability greater than $\frac{1}{2}$ are correctly classified. While this method yields impressive results, there is no guarantee that the confidence levels for correctly and incorrectly classified samples will not overlap in the interval $[\frac{1}{C}, \frac{1}{2}]$. As a result, the algorithm may assign different confidences to the same samples across multiple iterations, leading to a good overall performance at the cost of one or more classes being rejected more than necessary in some iterations if their samples are assigned too low a confidence. This can be observed in Figure 1, where these methods exhibit very low coverage with high associated risk for the class deer in CIFAR-10. Such a phenomenon is repeated over different classes in the set of experiments with 10 different initialization seeds considered in Figure 4, where ConfidNet exhibits many outliers in both coverage and risk.

**DeepGamblers** (DG): Liu et al. (2019) implements the rejection option for a predictor $f$ by optimizing $\mathcal{L}_{DG}(h; \mathcal{D}_n) = -\frac{1}{n} \sum_{i=1}^{n} \log(h_{y_i}(\mathbf{x}_i) o_i + h_{C+1}(\mathbf{x}_i))$, where $h_{y_i}$ represents the confidence associated to the correct class $y_i \in \mathcal{Y}$ for the input sample $\mathbf{x}_i \in \mathcal{X}$, $h_{C+1}(\cdot)$ is the model's confidence for the extra class that represents the selection and the coefficient $o$ is the payoff associated to accepting a decision. In a nutshell, the function above corresponds to the classical cross-entropy for $o_i = 1$ and $h_{C+1}(\cdot) = 0$. The training algorithm requires a warm-up period during which the weights are updated by optimizing the classic cross-entropy loss for about one-third of the total training iterations. After that, the loss above replaces the standard cross-entropy and the model is trained to learn when to accept/reject decisions. As by admission of the authors themselves, the cost term $o_i$, which is a hyper-parameter that requires tuning through the information conveyed by extra samples, is a key component of the algorithm. In particular, a lower $o$ causes the model to learn to reject better, but with a larger variance. Due to the lack of any indication on how to choose the right value for $o$, and since we did not use validation to be fair to the other frameworks, in our experiments, we maintained a fixed value $o = 2.2$ for CIFAR-10, $o = 2.6$ for SVHN reported as indicated in the original paper Liu et al. (2019), and $o = 2.0$ for CIFAR-100 which is the suggested default value. Figure 1 exhibits subpar coverage and high risk for this method when considering the class bird in CIFAR-10. Non-negligible variance, although lower than w.r.t. other competitors, is consistently reported across the experiments with 10 different initialization seeds reported in the box-plots in Figure 4.

**Self-Adaptive Training** (SAT): although Huang et al. (2020) primarily deals with the problem of generalization improvement under the assumption of potentially corrupted training data, the authors show how their method can be adapted to the problem of selective classification by adding an extra class to represent the rejection (as in Liu et al. (2019)) directly using the model prediction as a signal for learning abstention. The model is initially trained with the standard cross-entropy loss function during a warm-up set of epochs (60 in the basic algorithm reported in the paper). Then the cross-entropy loss is replaced by $\mathcal{L}_{SAT}(h; \mathcal{D}_n) = -\frac{1}{n} \sum_{i=1}^{n} [t_{i,y_i} \log(h_{y_i}(\mathbf{x}_i)) + (1 - t_{i,y_i}) \log(h_{C+1}(\mathbf{x}_i))]$. In this equation, the subscript $y_i$ is the index of the correct class for the input $\mathbf{x}_i$ and $\mathbf{t}_i$ is a convex combination defined as $\mathbf{t}_i = \alpha \times \mathbf{y}_i + (1 - \alpha) \times h(\mathbf{x}_i)$, where $\mathbf{y}_i$ is the one-hot encoded version of the true label for $\mathbf{x}_i$. According to the basic algorithm reported in the paper, $\alpha = 0.9$. Clearly a small $t_{i,y_i}$ reveals low confidence and enforces the selector to reject the decision. If $t_{i,y_i}$ is close to one, abstention becomes unlikely, and the function above recovers the standard cross-entropy. This framework achieves favorable results across the independent experiments summarized by the plot in Figure 4. However, the role played by a non-optimized number of warm-up epochs or value of $\alpha$ necessarily affects the value of $\mathbf{t}_i$, which, in turn, may cause poor performance as reported in Figure 1 for the class "truck" in CIFAR-10.

## 4 Optimal rejection option is achieved by the optimal misclassification detector

In this section, we establish the **relationship between the rejection option and misclassification detection for any selector**. We start by mathematically formalizing the optimal risk of the oracle (optimal Bayes) selector. We then show that, for a desired coverage, the optimal risk can be achieved by the optimal misclassification detector by proving that the optimization of this problem leads to the minimization of risk as defined in Equation (3). While Chow (1970) only takes into account the oracle detector, we consider a recently developed misclassification detection framework (Granese et al., 2021) that has been shown to outperform previous methods, and we argue for its use as a rejection/acceptance selector and an estimator of the true probability of error. Indeed, we believe that by showing the connection between selective risk and error probability, we shed light on the strong tightness between the two problems, bridging the gap between

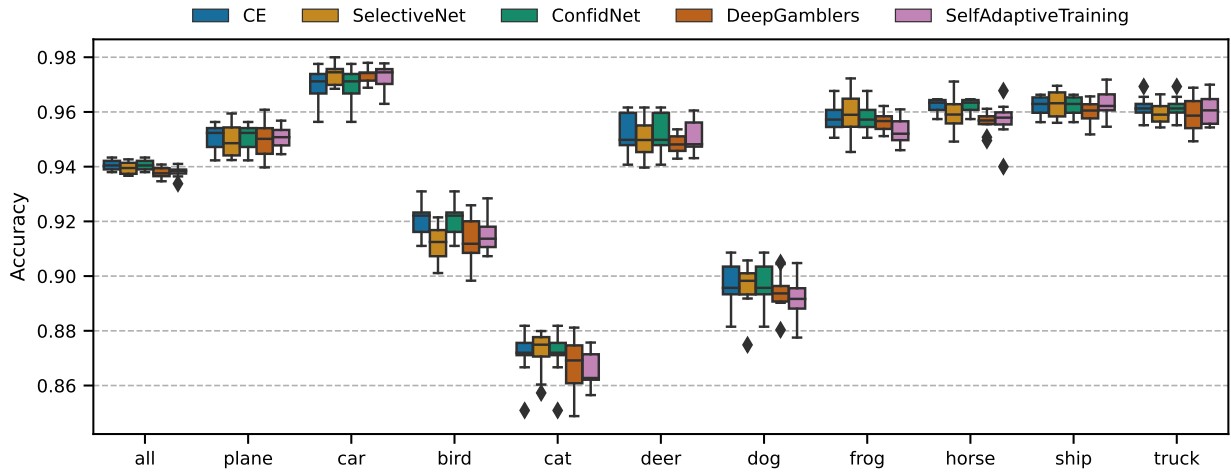

Figure 2: Global and class-wise accuracy for a VGG-16 model trained on CIFAR-10 with a cross-entropy (CE) loss and with losses defined on state-of-the-art training-based rejection option methods over 10 different runs.

the two communities. One of the key advantages of this method is that it relies on a state-of-the-art classifier trained using classical cross-entropy loss optimization, resulting in models that converge to similar final performance despite being randomly initialized. Our conjecture is that, as a consequence of this observed behavior, **the acceptance/rejection selection for these models within our proposed framework has lower risk and coverage variance compared to state-of-the-art methods**.

### 4.1 Preliminaries

In this subsection, we recall the concept of misclassification event. Let us consider a discrete r.v. expressed by $E \triangleq \mathbb{1}[f(\mathbf{X}) \neq Y]$, i.e., the *misclassification* event is denoted by $\{E = 1\} \equiv \{(\mathbf{x}, y) \in \mathcal{X} \times \mathcal{Y} : f(\mathbf{x}) \neq y\}$. We can express the joint probability density function $p_{XY}$ as a mixture:

$$p_{XY}(\mathbf{x}, y) = p_{XY|E}(\mathbf{x}, y | E = 1) P_E(1) + p_{XY|E}(\mathbf{x}, y | E = 0) P_E(0). \tag{4}$$

By taking the marginal of Equation (4) over $Y$, we obtain:

$$p_X(\mathbf{x}) = p_{X|E}(\mathbf{x}|1) P_E(1) + p_{X|E}(\mathbf{x}|0) P_E(0), \tag{5}$$

where $p_{X|E}(\mathbf{x}|1)$ denotes the pdf truncated to the error event and $p_{X|E}(\mathbf{x}|0)$ the pdf truncated to the event of correct classification.

The *probability of classification error* is finally defined as:

$$\text{Pe}(\mathbf{x}) \triangleq P_{E|X}(1|\mathbf{x}) = 1 - P_{Y|X}\left(f_{\mathcal{D}_n}(\mathbf{x}) \mid \mathbf{x}\right). \tag{6}$$

### 4.2 From selective risk to misclassification detection

Let us recall the risk definition from El-Yaniv & Wiener (2010) with the standard 0/1 loss as selector S. We will expand the selective model framework outlined in Section 3 by rewriting the risk in terms of the underlying probability distribution $p_X$ and the mixture model proposed in Equation (5):

$$r(f, S) \triangleq \frac{\mathbb{E}_{XY}[\mathbb{1}_{[f_{\mathcal{D}_n}(\mathbf{X}) \neq Y]} S(\mathbf{X})]}{\mathbb{E}_X[S(\mathbf{X})]} = \frac{\mathbb{E}_{X|E=1}[S(\mathbf{X})|E = 1]}{\mathbb{E}_{X|E=1}[S(\mathbf{X})|E = 1] + \beta \cdot \mathbb{E}_{X|E=0}[S(\mathbf{X})|E = 0]}, \tag{7}$$

where $\beta \triangleq \frac{P_E(0)}{P_E(1)} = \frac{\text{Accuracy}}{1 - \text{Accuracy}}$ is a constant[2]. We relegate the derivation of Equation (7) to the Appendix 8.1.

**Definition 4.1** (Minimum selective risk)**.** For a given desired coverage $\lambda \in (0, 1]$, the minimum selective risk is defined as follows:

$$r^\star(f, \lambda) \triangleq \inf_{\substack{S:\mathcal{X} \to \{0,1\} \text{ s.t. } \mathbb{E}_X[S(\mathbf{X})] \geq \lambda}} r(f, S), \tag{8}$$

where the infimum is taking over all selectors that satisfy the coverage constraint and $r(f, S)$ is defined in (7).

The next proposition shows that the minimum selective risk in (8) can be expressed in terms of $P_I$, i.e., the Error of Type-I which indicates the rejection of a sample that would have been correctly classified, and $P_{II}$, i.e., the Error of Type-II which indicates the probability of accepting a sample that would have been misclassified. Formally, we define

$$P_I \triangleq 1 - \mathbb{E}_{X|E=0}[S(\mathbf{X})|E = 0], \tag{9}$$

and

$$P_{II} \triangleq \mathbb{E}_{X|E=1}[S(\mathbf{X})|E = 1]. \tag{10}$$

Then, according to Granese et al. (2021) (cf. Proposition 3.1 therein), an oracle that knows the involved probabilities can select the best trade-off between $P_I$ and $P_{II}$ to minimize the selective risk in Equation (8). Furthermore, we introduce the proposition below, which provides the optimal selector achieving the minimum selective risk in Equation (8).

**Proposition 4.2.** *For a given desired coverage $\lambda \in (0, 1]$, the minimum selective risk defined in Equation (8) can be expressed in terms of an optimization over all feasible error probabilities $P_I$ and $P_{II}$ as follows:*

$$r^\star(f, \lambda) = \inf\left\{ \frac{P_{II}}{P_{II} + \beta(1 - P_I)} : (P_I, P_{II}) \in \mathcal{R} \text{ s.t. } P_E(1)P_{II} + P_E(0)(1 - P_I) \geq \lambda \right\}, \tag{11}$$

*where the set of all feasible error probabilities $\mathcal{R}$ is given by*

$$\mathcal{R} \triangleq \left\{ (P_I, P_{II}) \in [0, 1]^2 \text{ s.t. } P_I + P_{II} \geq 1 - \|p_{X|E=1} - p_{X|E=0}\|_{TV} \right\}. \tag{12}$$

*Hence, the optimal selector achieving the minimum selective risk in (8) is given by*

$$S^*(\mathbf{x}; \gamma^\star) = \mathbb{1}\left[ \frac{\text{Pe}(\mathbf{x})}{1 - \text{Pe}(\mathbf{x})} \leq \gamma^\star \right], \tag{13}$$

*where $\gamma^\star$ is the optimal value minimizing (7) and satisfying the coverage constraint $\mathbb{E}_X[S^*(\mathbf{X}; \gamma^\star)] \geq \lambda$.*

The proof to Proposition 4.2 is relegated to the Appendix (see Section 8.2).

### 4.3 Gini selector

Since the true probability of error Pe is unknown and cannot be learned from samples, we will rely on an approximation. Let us define:

$$\text{Gini}(\mathbf{x}) \triangleq \sum_{y \in \mathcal{Y}} P_{\widehat{Y}|X}(y|\mathbf{x}) \Pr(\widehat{Y} \neq y|\mathbf{x}) = 1 - \sum_{y \in \mathcal{Y}} P_{\widehat{Y}|X}^2(y|\mathbf{x}). \tag{14}$$

Then, from Proposition 3.2 in Granese et al. (2021), it holds that

$$1 - \sqrt{1 - \text{Gini}(\mathbf{x})} - \Delta(\mathbf{x}) \leq \text{Pe}(\mathbf{x}) \leq \text{Gini}(\mathbf{x}) + \Delta(\mathbf{x}), \tag{15}$$

where $\Delta(\mathbf{x}) = 2\sqrt{2KL(P_{Y|X}(\cdot, \mathbf{x}) \| P_{\widehat{Y}|X}(\cdot, \mathbf{x}))}$. Thus, a practical selector is given by $S(\mathbf{x}) = \mathbb{1}_{[\text{Gini}(\mathbf{x}) \leq \gamma]}$, finally connecting Equation (7) to Equation (14).

---

[2]We only consider the derivation for classification models that are not perfectly accurate, as otherwise, the necessity for a rejection option would not be adequately justified.

**Definition 4.3** (Rejection option with Gini($\cdot$))**.**

$$(f_{\mathcal{D}_n}, \text{Gini}, \gamma)(\mathbf{x}) \triangleq \begin{cases} f_{\mathcal{D}_n}(\mathbf{x}) & \text{if } \text{Gini}(\mathbf{x}) \leq \gamma \\ \emptyset & \text{if } \text{Gini}(\mathbf{x}) > \gamma, \end{cases} \tag{16}$$

where $\gamma \in [0, 1]$ is the threshold parameter and $\emptyset$ indicates that $f_{\mathcal{D}_n}$ abstains from the prediction.

Interestingly, we found out that Equation (14) can be linked to a popular information theoretic measure which is the Rényi divergence. We reference Section 8.3 for a more in-depth analysis.

With Equation (7) through Equation (14), and Definition 4.3 we have established a strong link between Type-I and Type-II errors from the point of view of misclassification detection and risk in the context of selective classification. This justifies the use of Equation (16) for post-training methods to implement the rejection option. In addition, Figure 2 shows that classifiers based on standard cross-entropy loss at training time (e.g. the base classified in ConfidNet) show consistently high accuracy with low variance, suggesting that this should also be reflected in the selection mechanism that accepts/rejects decisions.

Inspired by this observation, and aware of the limitations exposed in Section 3, we plot the empirical cumulative density function of Gini and ConfidNet over 10 runs in Figure 3. We can see that the Gini selection score has less distributional variability compared to ConfidNet. Although we cannot claim that the proposed method will always obtain better results without further assumption on the probability distributions involved (Lee & Barber, 2021; Zhang et al., 2021), this observation supports our suggestion that the use of a pure cross-entropy training approach followed by a post-hoc rejection method leads to more consistent results on popular benchmarks, and may be beneficial when seeking robust and reliable results in the context of selective classification, which is confirmed experimentally in Figure 4 and Section 6.

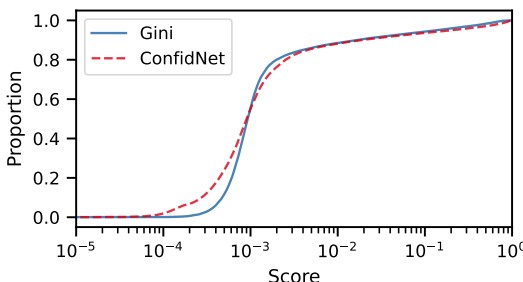

Figure 3: Empirical cdf of scores with 95% confidence interval for Gini and ConfidNet for 10 different runs with CIFAR-10 and VGG-16, showing an important gap in variance difference between a post-hoc and a training-based soft-selection score.

## 5 Experimental Design

We experimentally analyze SelectiveNet (cf. Geifman & El-Yaniv (2019)), ConfidNet (cf. Corbière et al. (2019)), DeepGambler (cf. Liu et al. (2019)), and SelfAdaptiveTraining (cf. Huang et al. (2020)) as training-based selective classification methods, compared to post-hoc methods such as MCDropout (cf. Gal & Ghahramani (2016)), ODIN (cf. Liang et al. (2017)), and our post-hoc method denoted Gini.

**Models and benchmark.** In our experiments, we consider three datasets: CIFAR-10, CIFAR-100 Krizhevsky (2009), and SVHN Netzer et al. (2011). For each of the train based methods, we train the underlying models using the entire training sets and following the training guidelines as reported in the corresponding papers. We train the standard classifier model, which we apply our Gini selector to, using the cross-entropy loss function and a stochastic gradient descent optimizer with momentum and learning rate scheduling for 300 epochs and a batch size of 64. To expand on previous research, we consider three neural network architectures: VGG-16, ResNet, and DenseNet, as well as the more challenging CIFAR-100 dataset. We conducted experiments with 10 different random seeds for each model, dataset, and method, and include error bars in our main results. Additionally, we also perform experiments on the larger ImageNet (Deng et al., 2009) dataset to further analyze the proposed post-hoc solution, which can be easily implemented using publicly available state-of-the-art model checkpoints. For Gini and ODIN (Liang et al., 2017), we also select the best parameters for the input perturbation magnitude and temperatures on the validation data. For MCDropout Gal & Ghahramani (2016), we average over 5 forward passes with a dropout probability of 0.3.

**Coverage calibration.** To ensure fair and comparable results, we standardize the procedure for coverage calibration across all methods, which was not always clear in previous works. We divide the test partition for

each of the three considered datasets into two subsets: one for calibration and another for evaluation. The calibration set corresponds to 10% of the original partition, chosen randomly for each of the ten seeds. The coverage calibration algorithm is outlined in Algorithm 1, where $\mathcal{D}_{m'}$ denotes the calibration dataset of size $m'$. We set target coverages $\tau$ from 0.50 to 1.00 with increments of 0.05. Intuitively, in order to guarantee the target coverage, we calculate scores for all samples in the calibration set and order them in ascending order. Then, we select the score value at index $\lceil \tau \cdot m' \rceil$ as threshold.

---

**Algorithm 1** Coverage calibration algorithm.

---

**Input:** Calibration set $\mathcal{D}_{m'} = \{(\mathbf{x}_i, y_i)\}_{i=1}^{m'}$, selector S, and target coverage $\tau \in [0, 1]$
SList = [ ]
**for** $i = 1$ **to** $m'$ **do**
  SList.append($S(\mathbf{x}_i)$)
**end for**
sort(SList, ascend=True)
**Return:** $\gamma^* = $ SList$[\lceil \tau \cdot m' \rceil]$

---

**Implementation details.** From an implementation point of view, post-hoc rejection option methods are the more resource-efficient and cost-effective solutions as they require fewer or no hyperparameters to be tuned and no architectural changes to the models. This is particularly beneficial in scenarios where it is difficult to collect additional samples for parameter optimization. For instance, SelectiveNet requires fitting a model for each target coverage. ConfidNet adds a significant overhead especially during inference, with an auxiliary confidence network with 1 million additional parameters. This overhead may limit some applications, e.g., ML applications on the edge (Murshed et al., 2022). DeepGamblers requires hyperparameter validation and warm-up with only CE loss. SelfAdaptiveTraining also requires warm-up with cross-entropy and may require tuning the momentum parameter, which is globally set to 0.9 in the original paper.

**Source code and computational resources.** In this study, we utilize a cluster of GPUs to train and evaluate the deep learning models, allowing for efficient parallelization of our experiments. Adhering to the principles of open science, we have made our code and trained models publicly available[3] to facilitate the reproducibility of our research. We hope that this benchmark will be useful to the research community and inspire further studies on rejection option.

## 6 Discussion

### 6.1 Main results

Gini and ODIN, despite their simplicity, exhibit comparable or superior performance across various target coverages for all three datasets. Specifically, for CIFAR-10, we see in Table 1 that the post-hocs methods achieve better performance on average from 50 to 100% coverage. The results on CIFAR-100, shown in Figure 5, demonstrate that the Gini selector outperforms the other methods and can achieve at most 5% risk with a 50% coverage, while the other methods fail to achieve so with a large gap. We believe this to be due, at least in part, to the limited availability of additional data to validate hyper-parameters for the training-based methods, which results in sub-optimal performance. This trend is also observed for the other models in the benchmark, for which we report extended results in Appendix 8.4. the proposed method outperforms, on average, all the state-of-the-art methods. On the SVHN dataset, the proposed method attains comparable performance for lower coverage levels and slightly worse performance on higher coverage rates. The ConfidNet and the SelfAdaptiveTraining frameworks perform poorly in this last benchmark.

In Figure 4, we compare the performance of the proposed method to that of competing methods over 10 seeds and classes individually. The image contains multiple box plots that depict the calibrated coverage and risk of the methods for each class for specific target coverage. The box plots clearly show that the proposed method has a more consistent performance across different classes and seeds, with fewer outliers. In contrast, the competing methods exhibit a higher degree of variability, with many outliers. The image

---

[3]Anonymized source code: `https://github.com/giniselector/giniselector`.

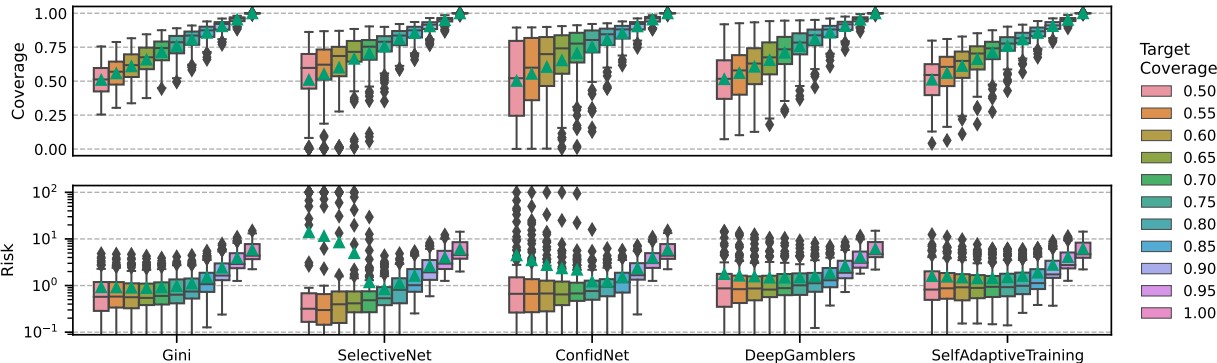

Figure 4: Box plots for the VGG-16 (CIFAR-10) benchmark comparing the coverage and risk of the proposed method to that of competing methods over 10 seeds and classes individually. The values for each boxplot are risk (below) and coverage (above) collected for each combination of target coverage value, class label, and initialization seed. The proposed method shows fewer outliers, indicating a more reliable performance. The green triangle marker indicates the average metric and the whiskers show the median.

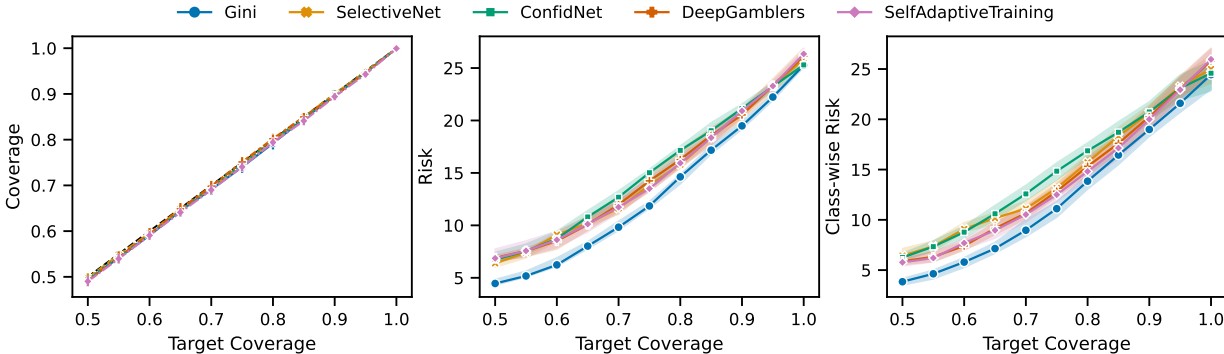

Figure 5: True coverage versus target coverage, risk versus target coverage, and class individual risk comparing methods on a VGG-16 (CIFAR-100) model over 10 different model initialization.

highlights the advantage of the proposed method in providing a more reliable and consistent performance across different classes and random initializations. These results demonstrate the effectiveness of the Gini selector in addressing the selective classification problem and its potential as a practical alternative to the training-based state-of-the-art. We provide similar plots in Section 8.4 for DenseNet and ResNet where similar trends are also observed.

## 6.2 Results on ImageNet

To study the relation between the accuracy and the risk linked to the rejection option on a large-scale problem, we set up the following experiment. We consider five off-the-shelf pre-trained ResNet models with different numbers of parameters and increasing accuracy (ResNet-18, ResNet-34, ResNet-50, ResNet-101, and ResNet-152) from Paszke et al. (2019); we evaluate the performance of the proposed Gini selector on the ILSVRC2012, or ImageNet-1K dataset (Deng et al., 2009) validation partition for each of the target coverages. We use 10% of this partition for coverage calibration and 90% for evaluation purposes. Figure 6 shows that the empirical risk of our method decreases with the accuracy of the base model on the same task. Of course, this has the cost of increasing the number of parameters. Generally, increasing the accuracy by scaling up the base model will decrease the risk for fixed coverage, especially in the high coverage regime. These results reassure the practicality of the proposed post-hoc selection method.

Table 1: Empirical selective risk (the lower the better given a fixed coverage) in percentage for the classification benchmark with VGG-16 for various target coverages over 10 runs. "Cov." stands for the target coverage and MCD, SN, CN, DG, and SAT stands for MCDropout, SelectiveClassification, ConfidNet, DeepGamblers, and SelfAdaptiveTraining, respectively.

| | Cov. | Gini (Ours) | ODIN | MCD | SN | CN | DG | SAT |
|---|---|---|---|---|---|---|---|---|
| **VGG-16 (CIFAR-10)** | 0.50 | $\mathbf{0.31}_{\pm 0.1}$ | $\mathbf{0.31}_{\pm 0.1}$ | $0.74_{\pm 0.1}$ | $0.36_{\pm 0.0}$ | $0.80_{\pm 0.2}$ | $1.10_{\pm 0.1}$ | $1.36_{\pm 0.2}$ |
| | 0.55 | $\mathbf{0.31}_{\pm 0.1}$ | $\mathbf{0.31}_{\pm 0.1}$ | $0.74_{\pm 0.1}$ | $0.41_{\pm 0.1}$ | $0.79_{\pm 0.1}$ | $1.12_{\pm 0.1}$ | $1.33_{\pm 0.1}$ |
| | 0.60 | $\mathbf{0.33}_{\pm 0.1}$ | $\mathbf{0.33}_{\pm 0.1}$ | $0.74_{\pm 0.1}$ | $0.48_{\pm 0.1}$ | $0.78_{\pm 0.1}$ | $1.18_{\pm 0.1}$ | $1.32_{\pm 0.1}$ |
| | 0.65 | $\mathbf{0.37}_{\pm 0.1}$ | $\mathbf{0.37}_{\pm 0.1}$ | $0.75_{\pm 0.1}$ | $0.56_{\pm 0.1}$ | $0.78_{\pm 0.1}$ | $1.22_{\pm 0.1}$ | $1.31_{\pm 0.1}$ |
| | 0.70 | $\mathbf{0.45}_{\pm 0.0}$ | $\mathbf{0.45}_{\pm 0.0}$ | $0.79_{\pm 0.1}$ | $0.62_{\pm 0.1}$ | $0.85_{\pm 0.1}$ | $1.28_{\pm 0.1}$ | $1.36_{\pm 0.1}$ |
| | 0.75 | $\mathbf{0.55}_{\pm 0.0}$ | $\mathbf{0.55}_{\pm 0.0}$ | $0.85_{\pm 0.1}$ | $0.72_{\pm 0.2}$ | $0.94_{\pm 0.1}$ | $1.36_{\pm 0.1}$ | $1.42_{\pm 0.1}$ |
| | 0.80 | $0.76_{\pm 0.0}$ | $\mathbf{0.75}_{\pm 0.0}$ | $0.96_{\pm 0.1}$ | $0.99_{\pm 0.1}$ | $1.05_{\pm 0.1}$ | $1.48_{\pm 0.1}$ | $1.53_{\pm 0.1}$ |
| | 0.85 | $\mathbf{1.21}_{\pm 0.1}$ | $\mathbf{1.21}_{\pm 0.1}$ | $1.33_{\pm 0.1}$ | $1.56_{\pm 0.0}$ | $1.33_{\pm 0.1}$ | $1.76_{\pm 0.2}$ | $1.80_{\pm 0.1}$ |
| | 0.90 | $\mathbf{2.12}_{\pm 0.2}$ | $\mathbf{2.12}_{\pm 0.2}$ | $2.24_{\pm 0.2}$ | $2.51_{\pm 0.4}$ | $2.20_{\pm 0.1}$ | $2.54_{\pm 0.2}$ | $2.58_{\pm 0.3}$ |
| | 0.95 | $\mathbf{3.59}_{\pm 0.2}$ | $3.63_{\pm 0.2}$ | $3.74_{\pm 0.2}$ | $3.86_{\pm 0.3}$ | $3.75_{\pm 0.3}$ | $3.97_{\pm 0.1}$ | $3.90_{\pm 0.4}$ |
| | 1.00 | $\mathbf{5.80}_{\pm 0.3}$ | $\mathbf{5.80}_{\pm 0.3}$ | $5.85_{\pm 0.3}$ | $5.96_{\pm 0.2}$ | $5.86_{\pm 0.2}$ | $6.28_{\pm 0.1}$ | $6.26_{\pm 0.2}$ |
| **VGG-16 (CIFAR-100)** | 0.50 | $\mathbf{3.78}_{\pm 0.3}$ | $3.79_{\pm 0.3}$ | $4.52_{\pm 0.3}$ | $6.45_{\pm 0.6}$ | $6.79_{\pm 0.7}$ | $6.60_{\pm 0.6}$ | $7.38_{\pm 0.7}$ |
| | 0.55 | $\mathbf{4.80}_{\pm 0.4}$ | $4.81_{\pm 0.4}$ | $5.32_{\pm 0.3}$ | $7.10_{\pm 0.4}$ | $7.63_{\pm 0.6}$ | $7.18_{\pm 0.6}$ | $7.95_{\pm 0.7}$ |
| | 0.60 | $\mathbf{6.09}_{\pm 0.3}$ | $\mathbf{6.09}_{\pm 0.3}$ | $6.43_{\pm 0.5}$ | $8.84_{\pm 0.6}$ | $8.83_{\pm 0.6}$ | $8.15_{\pm 0.8}$ | $8.86_{\pm 0.8}$ |
| | 0.65 | $\mathbf{7.87}_{\pm 0.4}$ | $\mathbf{7.87}_{\pm 0.5}$ | $8.06_{\pm 0.6}$ | $10.55_{\pm 0.5}$ | $10.67_{\pm 0.6}$ | $9.73_{\pm 0.8}$ | $10.27_{\pm 0.8}$ |
| | 0.70 | $\mathbf{9.45}_{\pm 0.5}$ | $\mathbf{9.45}_{\pm 0.5}$ | $9.65_{\pm 0.7}$ | $12.12_{\pm 1.1}$ | $12.64_{\pm 0.6}$ | $11.49_{\pm 0.6}$ | $11.99_{\pm 0.8}$ |
| | 0.75 | $\mathbf{11.66}_{\pm 0.6}$ | $11.68_{\pm 0.6}$ | $11.84_{\pm 0.7}$ | $13.50_{\pm 0.7}$ | $14.59_{\pm 0.9}$ | $13.73_{\pm 0.8}$ | $13.82_{\pm 0.8}$ |
| | 0.80 | $\mathbf{14.13}_{\pm 0.8}$ | $14.21_{\pm 0.8}$ | $14.40_{\pm 0.9}$ | $15.71_{\pm 0.3}$ | $16.93_{\pm 0.8}$ | $15.99_{\pm 0.9}$ | $16.19_{\pm 0.8}$ |
| | 0.85 | $\mathbf{16.59}_{\pm 0.7}$ | $16.75_{\pm 0.8}$ | $17.16_{\pm 1.1}$ | $18.57_{\pm 0.7}$ | $18.88_{\pm 0.8}$ | $18.11_{\pm 0.7}$ | $18.58_{\pm 0.8}$ |
| | 0.90 | $\mathbf{19.13}_{\pm 0.8}$ | $19.31_{\pm 0.7}$ | $19.76_{\pm 0.6}$ | $20.51_{\pm 0.6}$ | $21.18_{\pm 0.6}$ | $20.60_{\pm 0.8}$ | $20.93_{\pm 0.9}$ |
| | 0.95 | $\mathbf{22.08}_{\pm 0.3}$ | $22.12_{\pm 0.4}$ | $22.28_{\pm 0.3}$ | $23.39_{\pm 0.3}$ | $23.23_{\pm 0.3}$ | $23.10_{\pm 0.7}$ | $23.40_{\pm 0.7}$ |
| | 1.00 | $\mathbf{25.31}_{\pm 0.2}$ | $25.34_{\pm 0.2}$ | $25.37_{\pm 0.2}$ | $25.89_{\pm 0.4}$ | $25.44_{\pm 0.2}$ | $26.09_{\pm 0.5}$ | $26.44_{\pm 0.4}$ |
| **VGG-16 (SVHN)** | 0.50 | $\mathbf{0.34}_{\pm 0.0}$ | $\mathbf{0.34}_{\pm 0.0}$ | $0.39_{\pm 0.1}$ | $0.70_{\pm 0.2}$ | $2.07_{\pm 0.5}$ | $0.87_{\pm 0.1}$ | $16.72_{\pm 35.7}$ |
| | 0.55 | $\mathbf{0.37}_{\pm 0.0}$ | $\mathbf{0.37}_{\pm 0.0}$ | $0.41_{\pm 0.1}$ | $0.79_{\pm 0.2}$ | $2.21_{\pm 0.5}$ | $0.86_{\pm 0.1}$ | $16.71_{\pm 35.7}$ |
| | 0.60 | $\mathbf{0.42}_{\pm 0.0}$ | $\mathbf{0.42}_{\pm 0.0}$ | $0.44_{\pm 0.0}$ | $0.83_{\pm 0.3}$ | $2.40_{\pm 0.4}$ | $0.86_{\pm 0.1}$ | $16.71_{\pm 35.7}$ |
| | 0.65 | $\mathbf{0.47}_{\pm 0.0}$ | $\mathbf{0.47}_{\pm 0.0}$ | $0.49_{\pm 0.0}$ | $0.71_{\pm 0.1}$ | $2.60_{\pm 0.5}$ | $0.85_{\pm 0.1}$ | $16.71_{\pm 35.7}$ |
| | 0.70 | $\mathbf{0.53}_{\pm 0.0}$ | $\mathbf{0.53}_{\pm 0.0}$ | $0.57_{\pm 0.1}$ | $0.69_{\pm 0.1}$ | $2.79_{\pm 0.5}$ | $0.85_{\pm 0.1}$ | $16.72_{\pm 35.7}$ |
| | 0.75 | $\mathbf{0.61}_{\pm 0.0}$ | $\mathbf{0.61}_{\pm 0.0}$ | $0.66_{\pm 0.0}$ | $0.70_{\pm 0.1}$ | $2.99_{\pm 0.4}$ | $0.85_{\pm 0.1}$ | $16.72_{\pm 35.7}$ |
| | 0.80 | $0.74_{\pm 0.0}$ | $0.74_{\pm 0.0}$ | $0.81_{\pm 0.1}$ | $\mathbf{0.66}_{\pm 0.1}$ | $3.21_{\pm 0.4}$ | $0.89_{\pm 0.1}$ | $16.72_{\pm 35.7}$ |
| | 0.85 | $1.04_{\pm 0.1}$ | $1.04_{\pm 0.1}$ | $1.09_{\pm 0.1}$ | $0.92_{\pm 0.1}$ | $3.52_{\pm 0.3}$ | $\mathbf{0.94}_{\pm 0.1}$ | $16.78_{\pm 35.6}$ |
| | 0.90 | $1.66_{\pm 0.1}$ | $1.66_{\pm 0.1}$ | $1.70_{\pm 0.1}$ | $1.31_{\pm 0.1}$ | $3.93_{\pm 0.2}$ | $\mathbf{1.26}_{\pm 0.1}$ | $17.05_{\pm 35.5}$ |
| | 0.95 | $3.00_{\pm 0.2}$ | $3.01_{\pm 0.2}$ | $3.03_{\pm 0.2}$ | $2.61_{\pm 0.2}$ | $4.43_{\pm 0.1}$ | $\mathbf{2.31}_{\pm 0.2}$ | $17.92_{\pm 35.0}$ |
| | 1.00 | $5.25_{\pm 0.1}$ | $5.22_{\pm 0.1}$ | $5.28_{\pm 0.1}$ | $\mathbf{4.26}_{\pm 0.1}$ | $5.28_{\pm 0.1}$ | $4.56_{\pm 0.2}$ | $19.60_{\pm 34.0}$ |

### 6.3 On the impact of calibration on Gini selector

In this section, we investigate whether models with calibrated probability predictions help improve the rejection option capabilities of our method. In Table 2 we computed the AURC (area under the risk-coverage curve) for temperature one (Uncal. Gini) and for the optimal calibration temperature (Cal. Gini) on the validation set. We ran experiments over ten different initialization of the deep models of this work. Even though the ECE decreases significantly with calibration, the AURC remains equivalent. Hence, we showed post-hoc posterior probability calibration does not improve the rejection option.

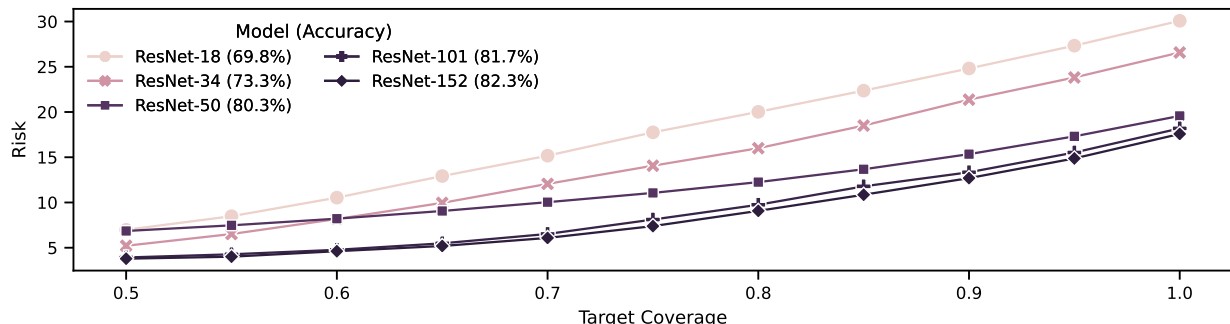

Figure 6: Risk-coverage curves for the Gini Selector on five ResNet models of different sizes and accuracies trained on ImageNet.

Table 2: Impact of model posterior probability calibration with temperature scaling on Gini selector method. The uncalibrated and the calibrated performances are in terms of average AURC (lower is better) and one standard deviation over ten different seeds in parenthesis. $ECE_1$ stands for the expected calibration error with temperature one and $ECE_T$ the expected calibration error with the optimal calibration temperature on the validation set.

| Architecture | Dataset | $ECE_1$ | $ECE_T$ | Uncal. Gini | Cal. Gini |
|---|---|---|---|---|---|
| VGG-16 | CIFAR-10 | 0.043 (0.001) | 0.012 (0.001) | 0.769 (0.053) | 0.824 (0.059) |
| | CIFAR-100 | 0.153 (0.003) | 0.035 (0.002) | 6.649 (0.107) | 6.749 (0.130) |
| | SVHN | 0.042 (0.001) | 0.004 (0.001) | 0.606 (0.025) | 0.624 (0.024) |
| DenseNet-121 | CIFAR-10 | 0.031 (0.001) | 0.006 (0.001) | 0.694 (0.028) | 0.716 (0.032) |
| | CIFAR-100 | 0.094 (0.003) | 0.015 (0.003) | 7.112 (0.161) | 7.196 (0.175) |
| | SVHN | 0.026 (0.001) | 0.005 (0.001) | 0.539 (0.030) | 0.553 (0.032) |
| ResNet-34 | CIFAR-10 | 0.030 (0.001) | 0.009 (0.000) | 0.443 (0.032) | 0.461 (0.033) |
| | CIFAR-100 | 0.060 (0.009) | 0.041 (0.002) | 4.886 (0.157) | 4.932 (0.161) |
| | SVHN | 0.024 (0.001) | 0.006 (0.001) | 0.473 (0.029) | 0.475 (0.029) |

## 7 Conclusion

This paper investigates the issue of the large variance in risk and coverage across classes for train-based models for selective classification. We provide a conjecture on the causes of this variation, presenting empirical evidence to support our findings through results obtained from multiple models and benchmarks. Furthermore, we establish a mathematical link between minimizing risk in selective classification and minimizing errors in misclassification detection. Our proposed solution offers a practical way to incorporate the rejection option for standard pre-trained classifiers. Perhaps, the main takeaway is that there is no free lunch when training confidence ranking functions, as their impressive global performance might come at the cost of unwanted behaviors across subgroups. Thus, we believe that this problem is open and that our results will encourage further research in the area.

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

# 8 Appendix

## 8.1 Full derivation of Equation (7)

In this section, we include a more thorough derivation of the link between the selective risk and the probability of errors of Type-I and Type-II:

$$r(f, \mathrm{S}) \triangleq \frac{\mathbb{E}_{XY}[\mathbb{1}_{[f_{\mathcal{D}_n}(\mathbf{X}) \neq Y]} \mathrm{S}(\mathbf{X})]}{\mathbb{E}_X[\mathrm{S}(\mathbf{X})]} \tag{17}$$

$$= \frac{P_E(1)\mathbb{E}_{XY|E=1}[\mathbb{1}_{[f_{\mathcal{D}_n}(\mathbf{X}) \neq Y]} \mathrm{S}(\mathbf{X})|E=1] + P_E(0)\mathbb{E}_{XY|E=0}[\mathbb{1}_{[f_{\mathcal{D}_n}(\mathbf{X}) \neq Y]} \mathrm{S}(\mathbf{X})|E=0]}{\mathbb{E}_X[\mathrm{S}(\mathbf{X})]} \tag{18}$$

$$= \frac{P_E(1)\mathbb{E}_{XY|E=1}[\mathbb{1}_{[f_{\mathcal{D}_n}(\mathbf{X}) \neq Y]} \mathrm{S}(\mathbf{X})|E=1]}{\mathbb{E}_X[\mathrm{S}(\mathbf{X})]} \tag{19}$$

$$= \frac{P_E(1)\mathbb{E}_{XY|E=1}[\mathrm{S}(\mathbf{X})|E=1]}{\mathbb{E}_X[\mathrm{S}(\mathbf{X})]} \tag{20}$$

$$= \frac{P_E(1)\mathbb{E}_{X|E=1}[\mathrm{S}(\mathbf{X})|E=1]}{P_E(1)\mathbb{E}_{X|E=1}[\mathrm{S}(\mathbf{X})|E=1] + P_E(0)\mathbb{E}_{X|E=0}[\mathrm{S}(\mathbf{X})|E=0]} \tag{21}$$

$$= \frac{\mathbb{E}_{X|E=1}[\mathrm{S}(\mathbf{X})|E=1]}{\mathbb{E}_{X|E=1}[\mathrm{S}(\mathbf{X})|E=1] + \frac{P_E(0)}{P_E(1)}\mathbb{E}_{X|E=0}[\mathrm{S}(\mathbf{X})|E=0]}, \tag{22}$$

where the desired identity follows from the definition of $\beta$.

## 8.2 Proof of Proposition 4.2

First, we establish a mathematical connection between the problem of detecting misclassifications and the problem of risk minimization for selective classification, by proving that the best (Oracle) misclassification detector corresponds to the selector which attains minimal risk for selective classification.
By considering the defintion in Equations (9) and (10), Equation (8) can be rewritten as

$$r^\star(f, \lambda) \triangleq \inf_{\mathrm{S}:\mathcal{X} \to \{0,1\} \text{ s.t. } \mathbb{E}_X[\mathrm{S}(\mathbf{X})] \geq \lambda} r(f, \mathrm{S})$$

$$= \inf \left\{ \frac{P_{\mathrm{II}}}{P_{\mathrm{II}} + \beta(1 - P_{\mathrm{I}})} : (P_{\mathrm{I}}, P_{\mathrm{II}}) \in \mathcal{R} \text{ s.t. } P_E(1)P_{\mathrm{II}} + P_E(0)(1 - P_{\mathrm{I}}) \geq \lambda \right\},$$

where, following Proposition 3.1 in Granese et al. (2021), we know that for any selector $S(\cdot)$, all corresponding feasible error probabilities must belong to

$$\mathcal{R} \triangleq \left\{ (P_{\mathrm{I}}, P_{\mathrm{II}}) \in [0,1]^2 \text{s.t. } P_{\mathrm{I}} + P_{\mathrm{II}} \geq 1 - \|p_{X|E=1} - p_{X|E=0}\|_{\mathrm{TV}} \right\},$$

and, among the feasible solutions in $\mathcal{R}$, we only consider those satisfying the coverage constraint $\lambda$ in Equation (11). The risk minimization over the tuples $(P_I, P_{\mathrm{II}}) \in \mathcal{R}$ highlights the fact that in order to minimize the risk one should try to jointly optimize over both probabilities.
Proposition 3.1 Granese et al. (2021) also states that the minimum feasible probability trade-offs $(P_{\mathrm{I}}, P_{\mathrm{II}})$ are achieved by letting

$$\mathrm{S}^*(\mathbf{x}; \gamma) = \mathbb{1}\left[ \frac{\mathrm{Pe}(\mathbf{x})}{1 - \mathrm{Pe}(\mathbf{x})} \leq \gamma \right],$$

for any arbitrary $\gamma \geq 0$.
This leads to feasible achievable probabilities $(P_{\mathrm{I}}, P_{\mathrm{II}})$ that achieve the equality:

$$P_{\mathrm{I}} + P_{\mathrm{II}} = 1 - \|p_{X|E=1} - p_{X|E=0}\|_{\mathrm{TV}},$$

as shown in Proposition 3.1 Granese et al. (2021).

Finally, by selecting a value for $\gamma$ in agreement with the constraint $\lambda$ on the coverage, we have that

$$S^*(\mathbf{x}; \gamma^\star) = \mathbb{1}\left[\frac{\text{Pe}(\mathbf{x})}{1 - \text{Pe}(\mathbf{x})} \leq \gamma^\star\right],$$

where $\gamma^\star$ is the optimal value minimizing (7) and satisfying the coverage constraint $\mathbb{E}_X[S^*(\mathbf{X}; \gamma^\star)] \geq \lambda$. This concludes the proof of the proposition. $\square$

*Remark* 8.1. We would like to remark that without the constraint on the coverage, the risk in Equation (11) can be rewritten as

$$\frac{P_{\text{II}}}{P_{\text{II}} + \beta\left(P_{\text{II}} + \|p_{X|E=1} - p_{X|E=0}\|_{\text{TV}}\right)}$$

and can be minimized for $P_{\text{II}} \to 0$. However, the constraint imposed on the coverage implies

$$P_{\text{II}} + \beta(1 - P_{\text{I}}) > \widehat{\lambda};$$

for $\widehat{\lambda} = \frac{\lambda}{P_E(1)}$. This means that, given a constraint $\lambda$ on the coverage,

$$P_{\text{II}} > \frac{\widehat{\lambda} - \beta\|p_{X|E=1} - p_{X|E=0}\|_{\text{TV}}}{1 + \beta},$$

which implies that $P_{\text{II}}$ cannot be arbitrarily minimized without taking into account the constraint $\lambda$ on the coverage.

### 8.3 On the derivation of Gini$(\cdot)$ from the Rényi divergence

Let us define the Rényi divergence, as:

$$D_\alpha\left(P_{\widehat{Y}|X}(\cdot, \mathbf{x})\|Q_Y\right) \doteq \frac{1}{\alpha - 1}\log\left(\sum_{y \in \mathcal{Y}}\left(P_{\widehat{Y}|X}^\alpha(y|\mathbf{x})Q_Y^{(1-\alpha)}(y)\right)\right), \tag{23}$$

where $P_{\widehat{Y}|X}(\cdot, \mathbf{x})$ is the model soft-distribution for a fixed input sample $\mathbf{x}$, and $Q_Y$ is a distribution over the labels set $\mathcal{Y}$. By fixing $\alpha = 2$, Equation (23) becomes

$$D_2\left(P_{\widehat{Y}|X}(\cdot, \mathbf{x})\|Q_Y\right) = \log\left(\sum_{y \in \mathcal{Y}}\left(\frac{P_{\widehat{Y}|X}^2(y|\mathbf{x})}{Q_Y(y)}\right)\right). \tag{24}$$

Let us now take a closer look at the argument of the logarithm. Let us fix the reference distribution $Q_Y$ as a *uniform distribution* over the classes, i.e. $Q_Y = q$, $\forall y \in \mathcal{Y}$. Then, the argument of the logarithm writes

$$\frac{1}{q}\sum_{y \in \mathcal{Y}}\left(P_{\widehat{Y}|X}^2(y|\mathbf{x})\right) \tag{25}$$

which corresponds to $1 - \text{Gini}(\mathbf{x})$ multiplied by a constant.

### 8.4 Additional Results

In this section, we present additional tables and plots that supplement the analysis we performed in the main manuscript.

Table 3: Empirical selective risk in percentage for the classification benchmark with DenseNet-121 for various target coverages over 10 runs.

| | Cov. | Gini | ODIN | MCD | SN | CN | DG | SAT |
|---|---|---|---|---|---|---|---|---|
| DenseNet-121 (CIFAR-10) | 0.50 | $0.22_{\pm 0.1}$ | $0.22_{\pm 0.1}$ | $0.36_{\pm 0.1}$ | $0.33_{\pm 0.1}$ | $0.37_{\pm 0.1}$ | $0.94_{\pm 0.1}$ | $0.92_{\pm 0.3}$ |
| | 0.55 | $0.25_{\pm 0.1}$ | $0.25_{\pm 0.1}$ | $0.37_{\pm 0.1}$ | $0.40_{\pm 0.1}$ | $0.42_{\pm 0.1}$ | $0.97_{\pm 0.1}$ | $0.95_{\pm 0.3}$ |
| | 0.60 | $0.30_{\pm 0.1}$ | $0.30_{\pm 0.1}$ | $0.42_{\pm 0.1}$ | $0.39_{\pm 0.1}$ | $0.47_{\pm 0.1}$ | $1.01_{\pm 0.1}$ | $0.98_{\pm 0.3}$ |
| | 0.65 | $0.34_{\pm 0.1}$ | $0.34_{\pm 0.1}$ | $0.52_{\pm 0.1}$ | $0.58_{\pm 0.1}$ | $0.53_{\pm 0.1}$ | $1.08_{\pm 0.1}$ | $1.02_{\pm 0.3}$ |
| | 0.70 | $0.46_{\pm 0.1}$ | $0.46_{\pm 0.1}$ | $0.63_{\pm 0.1}$ | $0.62_{\pm 0.1}$ | $0.66_{\pm 0.1}$ | $1.10_{\pm 0.1}$ | $1.08_{\pm 0.3}$ |
| | 0.75 | $0.64_{\pm 0.0}$ | $0.64_{\pm 0.0}$ | $0.76_{\pm 0.1}$ | $0.71_{\pm 0.1}$ | $0.79_{\pm 0.1}$ | $1.17_{\pm 0.0}$ | $1.19_{\pm 0.3}$ |
| | 0.80 | $0.88_{\pm 0.1}$ | $0.88_{\pm 0.1}$ | $1.00_{\pm 0.1}$ | $0.91_{\pm 0.2}$ | $1.02_{\pm 0.1}$ | $1.33_{\pm 0.0}$ | $1.31_{\pm 0.3}$ |
| | 0.85 | $1.29_{\pm 0.1}$ | $1.29_{\pm 0.1}$ | $1.38_{\pm 0.1}$ | $1.33_{\pm 0.1}$ | $1.34_{\pm 0.1}$ | $1.58_{\pm 0.1}$ | $1.49_{\pm 0.2}$ |
| | 0.90 | $2.12_{\pm 0.1}$ | $2.13_{\pm 0.1}$ | $2.22_{\pm 0.2}$ | $1.99_{\pm 0.3}$ | $2.22_{\pm 0.1}$ | $2.11_{\pm 0.1}$ | $2.09_{\pm 0.2}$ |
| | 0.95 | $3.46_{\pm 0.2}$ | $3.54_{\pm 0.3}$ | $3.57_{\pm 0.3}$ | $3.41_{\pm 0.2}$ | $3.55_{\pm 0.3}$ | $3.20_{\pm 0.3}$ | $3.19_{\pm 0.3}$ |
| | 1.00 | $5.78_{\pm 0.1}$ | $5.79_{\pm 0.1}$ | $5.85_{\pm 0.1}$ | $5.27_{\pm 0.1}$ | $5.85_{\pm 0.1}$ | $5.22_{\pm 0.2}$ | $5.30_{\pm 0.3}$ |
| DenseNet-121 (CIFAR-100) | 0.50 | $4.43_{\pm 0.5}$ | $4.43_{\pm 0.5}$ | $4.50_{\pm 0.5}$ | $7.85_{\pm 0.9}$ | $4.54_{\pm 0.5}$ | $7.82_{\pm 0.6}$ | $7.11_{\pm 1.3}$ |
| | 0.55 | $5.67_{\pm 0.4}$ | $5.70_{\pm 0.5}$ | $5.76_{\pm 0.6}$ | $9.33_{\pm 1.0}$ | $6.10_{\pm 0.6}$ | $8.91_{\pm 0.5}$ | $8.31_{\pm 1.2}$ |
| | 0.60 | $7.32_{\pm 0.6}$ | $7.34_{\pm 0.6}$ | $7.40_{\pm 0.6}$ | $9.85_{\pm 1.3}$ | $7.80_{\pm 0.7}$ | $10.10_{\pm 0.5}$ | $9.39_{\pm 1.3}$ |
| | 0.65 | $9.25_{\pm 0.8}$ | $9.27_{\pm 0.8}$ | $9.25_{\pm 0.9}$ | $10.63_{\pm 0.5}$ | $9.87_{\pm 1.0}$ | $11.35_{\pm 0.4}$ | $10.88_{\pm 1.3}$ |
| | 0.70 | $11.39_{\pm 0.8}$ | $11.38_{\pm 0.8}$ | $11.46_{\pm 0.8}$ | $12.63_{\pm 1.1}$ | $12.09_{\pm 0.8}$ | $12.65_{\pm 0.3}$ | $12.42_{\pm 1.2}$ |
| | 0.75 | $13.41_{\pm 0.7}$ | $13.42_{\pm 0.7}$ | $13.52_{\pm 0.8}$ | $13.95_{\pm 1.1}$ | $14.15_{\pm 0.8}$ | $14.18_{\pm 0.2}$ | $14.09_{\pm 1.0}$ |
| | 0.80 | $15.54_{\pm 0.6}$ | $15.69_{\pm 0.6}$ | $15.98_{\pm 0.5}$ | $16.23_{\pm 0.8}$ | $16.50_{\pm 0.7}$ | $15.85_{\pm 0.3}$ | $15.85_{\pm 0.7}$ |
| | 0.85 | $17.97_{\pm 0.5}$ | $18.05_{\pm 0.5}$ | $18.14_{\pm 0.7}$ | $17.94_{\pm 0.7}$ | $18.79_{\pm 0.6}$ | $17.80_{\pm 0.2}$ | $17.93_{\pm 0.5}$ |
| | 0.90 | $20.50_{\pm 0.5}$ | $20.58_{\pm 0.4}$ | $20.67_{\pm 0.4}$ | $20.25_{\pm 0.9}$ | $21.08_{\pm 0.3}$ | $19.69_{\pm 0.3}$ | $19.77_{\pm 0.5}$ |
| | 0.95 | $23.11_{\pm 0.4}$ | $23.06_{\pm 0.4}$ | $23.20_{\pm 0.4}$ | $22.32_{\pm 0.6}$ | $23.48_{\pm 0.3}$ | $22.00_{\pm 0.6}$ | $22.09_{\pm 0.4}$ |
| | 1.00 | $26.05_{\pm 0.3}$ | $26.02_{\pm 0.3}$ | $26.10_{\pm 0.3}$ | $24.54_{\pm 0.9}$ | $25.87_{\pm 0.3}$ | $24.28_{\pm 0.5}$ | $24.48_{\pm 0.3}$ |
| DenseNet-121 (SVHN) | 0.50 | $0.50_{\pm 0.0}$ | $0.50_{\pm 0.0}$ | $0.64_{\pm 0.1}$ | $0.65_{\pm 0.2}$ | $4.81_{\pm 1.2}$ | $0.74_{\pm 0.1}$ | $0.82_{\pm 0.1}$ |
| | 0.55 | $0.50_{\pm 0.0}$ | $0.50_{\pm 0.0}$ | $0.64_{\pm 0.0}$ | $0.68_{\pm 0.1}$ | $4.76_{\pm 1.1}$ | $0.76_{\pm 0.1}$ | $0.82_{\pm 0.1}$ |
| | 0.60 | $0.50_{\pm 0.0}$ | $0.50_{\pm 0.0}$ | $0.67_{\pm 0.1}$ | $0.63_{\pm 0.1}$ | $4.69_{\pm 1.0}$ | $0.79_{\pm 0.1}$ | $0.83_{\pm 0.1}$ |
| | 0.65 | $0.53_{\pm 0.0}$ | $0.53_{\pm 0.0}$ | $0.68_{\pm 0.1}$ | $0.66_{\pm 0.1}$ | $4.67_{\pm 0.9}$ | $0.81_{\pm 0.1}$ | $0.84_{\pm 0.1}$ |
| | 0.70 | $0.56_{\pm 0.0}$ | $0.56_{\pm 0.0}$ | $0.70_{\pm 0.1}$ | $0.74_{\pm 0.1}$ | $4.63_{\pm 0.7}$ | $0.85_{\pm 0.1}$ | $0.86_{\pm 0.1}$ |
| | 0.75 | $0.59_{\pm 0.1}$ | $0.59_{\pm 0.1}$ | $0.74_{\pm 0.1}$ | $0.77_{\pm 0.1}$ | $4.62_{\pm 0.6}$ | $0.92_{\pm 0.1}$ | $0.91_{\pm 0.1}$ |
| | 0.80 | $0.65_{\pm 0.1}$ | $0.65_{\pm 0.1}$ | $0.79_{\pm 0.1}$ | $0.89_{\pm 0.1}$ | $4.60_{\pm 0.5}$ | $1.01_{\pm 0.2}$ | $0.96_{\pm 0.1}$ |
| | 0.85 | $0.74_{\pm 0.1}$ | $0.74_{\pm 0.1}$ | $0.86_{\pm 0.1}$ | $0.91_{\pm 0.1}$ | $4.62_{\pm 0.4}$ | $1.14_{\pm 0.2}$ | $1.09_{\pm 0.1}$ |
| | 0.90 | $1.04_{\pm 0.1}$ | $1.04_{\pm 0.1}$ | $1.11_{\pm 0.1}$ | $1.22_{\pm 0.1}$ | $4.67_{\pm 0.3}$ | $1.40_{\pm 0.2}$ | $1.33_{\pm 0.1}$ |
| | 0.95 | $1.89_{\pm 0.1}$ | $1.90_{\pm 0.1}$ | $1.94_{\pm 0.1}$ | $1.98_{\pm 0.1}$ | $4.74_{\pm 0.2}$ | $2.08_{\pm 0.2}$ | $2.05_{\pm 0.2}$ |
| | 1.00 | $3.93_{\pm 0.1}$ | $3.93_{\pm 0.1}$ | $3.95_{\pm 0.1}$ | $3.99_{\pm 0.1}$ | $4.94_{\pm 0.1}$ | $3.93_{\pm 0.1}$ | $3.94_{\pm 0.1}$ |

Table 4: Empirical selective risk in percentage for the classification benchmark with ResNet-34 for various target coverages over 10 runs.

| | Cov. | Gini | ODIN | MCD | SN | CN | DG | SAT |
|---|---|---|---|---|---|---|---|---|
| **ResNet-34 (CIFAR-10)** | 0.50 | $0.11_{\pm 0.0}$ | $0.11_{\pm 0.0}$ | $0.30_{\pm 0.0}$ | $0.15_{\pm 0.1}$ | $0.39_{\pm 0.1}$ | $0.40_{\pm 0.0}$ | $0.67_{\pm 0.2}$ |
| | 0.55 | $0.12_{\pm 0.0}$ | $0.12_{\pm 0.0}$ | $0.31_{\pm 0.1}$ | $0.16_{\pm 0.0}$ | $0.39_{\pm 0.1}$ | $0.43_{\pm 0.1}$ | $0.69_{\pm 0.2}$ |
| | 0.60 | $0.14_{\pm 0.0}$ | $0.14_{\pm 0.0}$ | $0.31_{\pm 0.1}$ | $0.21_{\pm 0.1}$ | $0.42_{\pm 0.1}$ | $0.47_{\pm 0.1}$ | $0.69_{\pm 0.2}$ |
| | 0.65 | $0.18_{\pm 0.0}$ | $0.18_{\pm 0.0}$ | $0.36_{\pm 0.1}$ | $0.26_{\pm 0.0}$ | $0.44_{\pm 0.1}$ | $0.50_{\pm 0.1}$ | $0.69_{\pm 0.2}$ |
| | 0.70 | $0.22_{\pm 0.0}$ | $0.22_{\pm 0.0}$ | $0.38_{\pm 0.1}$ | $0.25_{\pm 0.0}$ | $0.46_{\pm 0.1}$ | $0.54_{\pm 0.0}$ | $0.70_{\pm 0.2}$ |
| | 0.75 | $0.28_{\pm 0.0}$ | $0.28_{\pm 0.0}$ | $0.45_{\pm 0.1}$ | $0.33_{\pm 0.0}$ | $0.51_{\pm 0.1}$ | $0.60_{\pm 0.1}$ | $0.74_{\pm 0.2}$ |
| | 0.80 | $0.41_{\pm 0.1}$ | $0.41_{\pm 0.1}$ | $0.53_{\pm 0.1}$ | $0.50_{\pm 0.1}$ | $0.58_{\pm 0.1}$ | $0.69_{\pm 0.1}$ | $0.86_{\pm 0.3}$ |
| | 0.85 | $0.62_{\pm 0.1}$ | $0.62_{\pm 0.1}$ | $0.70_{\pm 0.1}$ | $0.79_{\pm 0.1}$ | $0.74_{\pm 0.1}$ | $0.85_{\pm 0.1}$ | $1.02_{\pm 0.2}$ |
| | 0.90 | $1.22_{\pm 0.2}$ | $1.22_{\pm 0.2}$ | $1.28_{\pm 0.2}$ | $1.21_{\pm 0.3}$ | $1.28_{\pm 0.2}$ | $1.33_{\pm 0.1}$ | $1.43_{\pm 0.3}$ |
| | 0.95 | $2.31_{\pm 0.3}$ | $2.33_{\pm 0.3}$ | $2.41_{\pm 0.2}$ | $2.65_{\pm 0.5}$ | $2.38_{\pm 0.3}$ | $2.37_{\pm 0.1}$ | $2.41_{\pm 0.4}$ |
| | 1.00 | $4.39_{\pm 0.1}$ | $4.38_{\pm 0.1}$ | $4.47_{\pm 0.2}$ | $4.52_{\pm 0.2}$ | $4.45_{\pm 0.2}$ | $4.40_{\pm 0.1}$ | $4.34_{\pm 0.2}$ |
| **ResNet-34 (CIFAR-100)** | 0.50 | $2.27_{\pm 0.3}$ | $2.29_{\pm 0.3}$ | $2.68_{\pm 0.3}$ | $4.19_{\pm 0.5}$ | $3.60_{\pm 0.2}$ | $3.48_{\pm 0.2}$ | $3.82_{\pm 0.2}$ |
| | 0.55 | $2.91_{\pm 0.3}$ | $2.92_{\pm 0.3}$ | $3.26_{\pm 0.3}$ | $5.31_{\pm 0.8}$ | $4.60_{\pm 0.2}$ | $4.31_{\pm 0.2}$ | $4.80_{\pm 0.2}$ |
| | 0.60 | $4.00_{\pm 0.1}$ | $4.00_{\pm 0.1}$ | $4.16_{\pm 0.2}$ | $5.68_{\pm 0.5}$ | $5.95_{\pm 0.4}$ | $5.38_{\pm 0.2}$ | $5.84_{\pm 0.4}$ |
| | 0.65 | $5.16_{\pm 0.2}$ | $5.16_{\pm 0.2}$ | $5.34_{\pm 0.2}$ | $7.70_{\pm 0.4}$ | $7.16_{\pm 0.6}$ | $6.86_{\pm 0.2}$ | $7.07_{\pm 0.4}$ |
| | 0.70 | $6.60_{\pm 0.3}$ | $6.60_{\pm 0.3}$ | $6.71_{\pm 0.3}$ | $9.06_{\pm 0.4}$ | $8.92_{\pm 0.5}$ | $8.38_{\pm 0.5}$ | $8.60_{\pm 0.5}$ |
| | 0.75 | $8.47_{\pm 0.4}$ | $8.46_{\pm 0.4}$ | $8.53_{\pm 0.4}$ | $10.84_{\pm 0.5}$ | $10.64_{\pm 0.7}$ | $10.27_{\pm 0.4}$ | $10.37_{\pm 0.5}$ |
| | 0.80 | $10.76_{\pm 0.5}$ | $10.71_{\pm 0.5}$ | $10.85_{\pm 0.6}$ | $12.82_{\pm 0.6}$ | $12.87_{\pm 0.7}$ | $12.17_{\pm 0.3}$ | $12.16_{\pm 0.7}$ |
| | 0.85 | $13.28_{\pm 0.5}$ | $13.14_{\pm 0.4}$ | $13.36_{\pm 0.6}$ | $14.71_{\pm 0.5}$ | $14.78_{\pm 0.6}$ | $14.47_{\pm 0.4}$ | $14.48_{\pm 0.7}$ |
| | 0.90 | $15.66_{\pm 0.4}$ | $15.64_{\pm 0.4}$ | $15.66_{\pm 0.5}$ | $17.08_{\pm 0.6}$ | $16.68_{\pm 0.4}$ | $16.46_{\pm 0.3}$ | $16.80_{\pm 0.6}$ |
| | 0.95 | $18.21_{\pm 0.4}$ | $18.16_{\pm 0.4}$ | $18.31_{\pm 0.4}$ | $18.87_{\pm 0.4}$ | $18.78_{\pm 0.6}$ | $18.99_{\pm 0.7}$ | $19.46_{\pm 0.6}$ |
| | 1.00 | $20.90_{\pm 0.4}$ | $20.88_{\pm 0.4}$ | $20.97_{\pm 0.5}$ | $21.78_{\pm 0.4}$ | $20.97_{\pm 0.5}$ | $21.64_{\pm 0.5}$ | $21.64_{\pm 0.3}$ |
| **ResNet-34 (SVHN)** | 0.50 | $0.42_{\pm 0.0}$ | $0.42_{\pm 0.0}$ | $0.54_{\pm 0.1}$ | $0.52_{\pm 0.1}$ | $4.20_{\pm 0.4}$ | $0.60_{\pm 0.1}$ | $0.69_{\pm 0.2}$ |
| | 0.55 | $0.43_{\pm 0.0}$ | $0.43_{\pm 0.0}$ | $0.55_{\pm 0.1}$ | $0.59_{\pm 0.1}$ | $4.37_{\pm 0.3}$ | $0.60_{\pm 0.1}$ | $0.73_{\pm 0.3}$ |
| | 0.60 | $0.44_{\pm 0.0}$ | $0.44_{\pm 0.0}$ | $0.56_{\pm 0.1}$ | $0.56_{\pm 0.0}$ | $4.48_{\pm 0.3}$ | $0.59_{\pm 0.1}$ | $0.75_{\pm 0.3}$ |
| | 0.65 | $0.45_{\pm 0.0}$ | $0.45_{\pm 0.0}$ | $0.56_{\pm 0.1}$ | $0.53_{\pm 0.1}$ | $4.56_{\pm 0.3}$ | $0.59_{\pm 0.1}$ | $0.79_{\pm 0.4}$ |
| | 0.70 | $0.46_{\pm 0.0}$ | $0.46_{\pm 0.0}$ | $0.57_{\pm 0.1}$ | $0.53_{\pm 0.0}$ | $4.59_{\pm 0.3}$ | $0.60_{\pm 0.1}$ | $0.82_{\pm 0.4}$ |
| | 0.75 | $0.47_{\pm 0.0}$ | $0.47_{\pm 0.0}$ | $0.59_{\pm 0.1}$ | $0.53_{\pm 0.1}$ | $4.63_{\pm 0.2}$ | $0.61_{\pm 0.1}$ | $0.85_{\pm 0.5}$ |
| | 0.80 | $0.51_{\pm 0.0}$ | $0.51_{\pm 0.0}$ | $0.62_{\pm 0.1}$ | $0.57_{\pm 0.0}$ | $4.70_{\pm 0.2}$ | $0.63_{\pm 0.1}$ | $0.91_{\pm 0.5}$ |
| | 0.85 | $0.66_{\pm 0.0}$ | $0.66_{\pm 0.0}$ | $0.70_{\pm 0.0}$ | $0.66_{\pm 0.1}$ | $4.75_{\pm 0.2}$ | $0.69_{\pm 0.0}$ | $1.01_{\pm 0.6}$ |
| | 0.90 | $0.96_{\pm 0.1}$ | $0.96_{\pm 0.1}$ | $1.00_{\pm 0.1}$ | $0.96_{\pm 0.1}$ | $4.81_{\pm 0.2}$ | $0.98_{\pm 0.1}$ | $1.27_{\pm 0.5}$ |
| | 0.95 | $1.75_{\pm 0.2}$ | $1.75_{\pm 0.2}$ | $1.76_{\pm 0.2}$ | $2.25_{\pm 0.3}$ | $4.84_{\pm 0.1}$ | $1.78_{\pm 0.2}$ | $1.92_{\pm 0.2}$ |
| | 1.00 | $3.86_{\pm 0.2}$ | $3.84_{\pm 0.2}$ | $3.87_{\pm 0.2}$ | $3.79_{\pm 0.1}$ | $5.01_{\pm 0.1}$ | $3.86_{\pm 0.1}$ | $3.86_{\pm 0.1}$ |

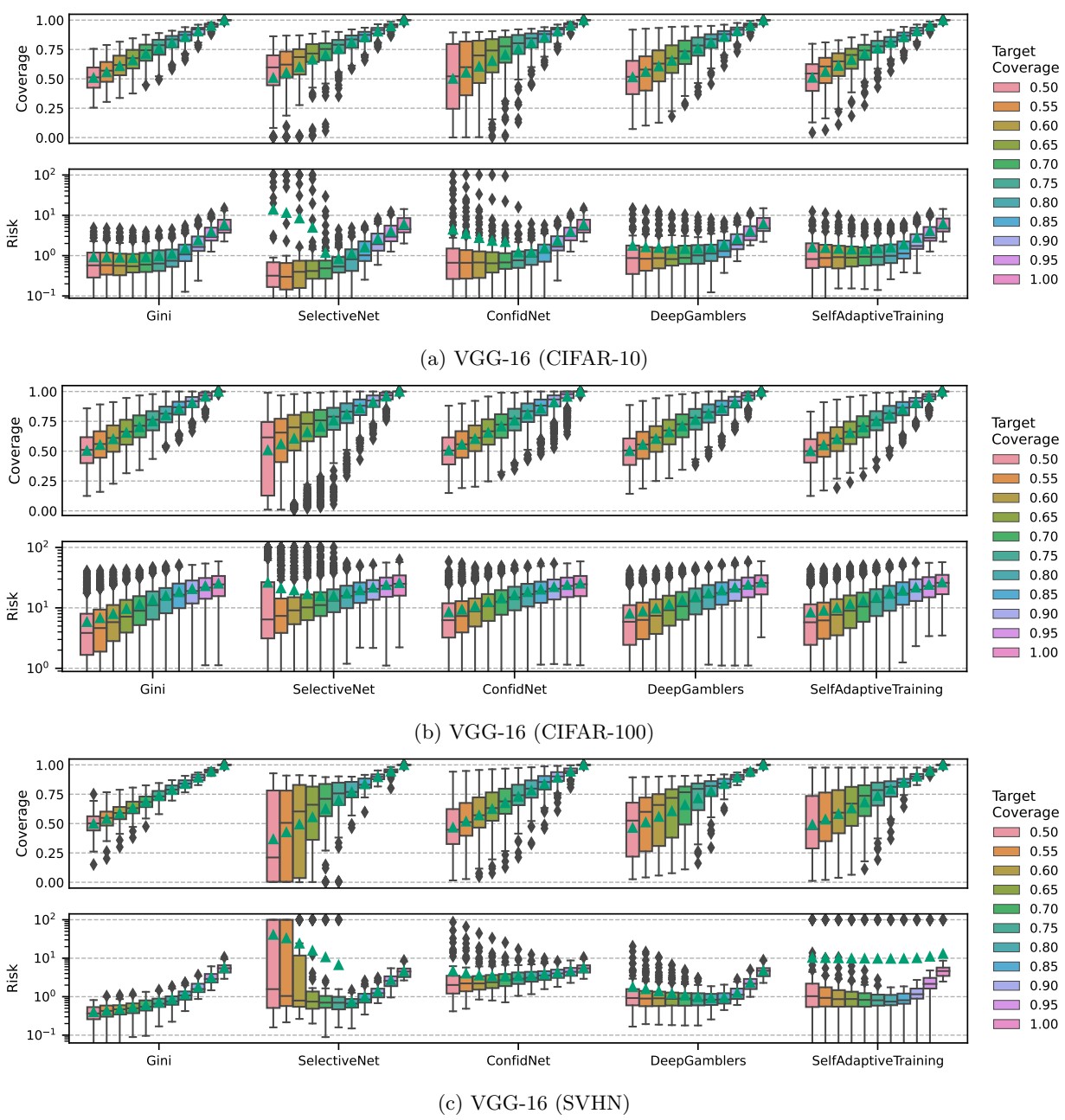

(a) VGG-16 (CIFAR-10)

(b) VGG-16 (CIFAR-100)

(c) VGG-16 (SVHN)

Figure 7: Class-wise calibrated coverage and risk versus target coverage for a VGG-16 model.

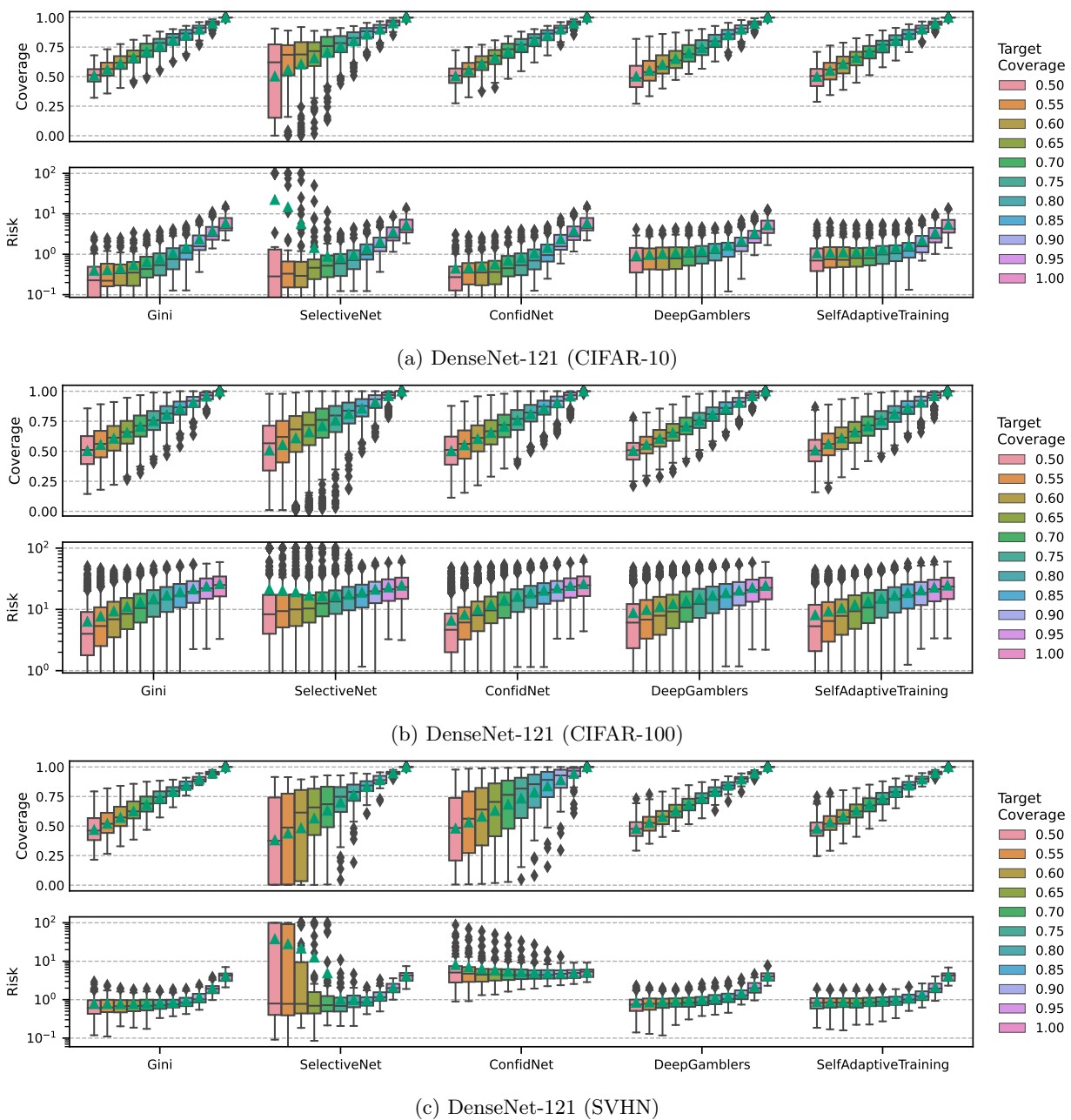

(a) DenseNet-121 (CIFAR-10)

(b) DenseNet-121 (CIFAR-100)

(c) DenseNet-121 (SVHN)

Figure 8: Class-wise calibrated coverage and risk versus target coverage for a DenseNet-121 model.

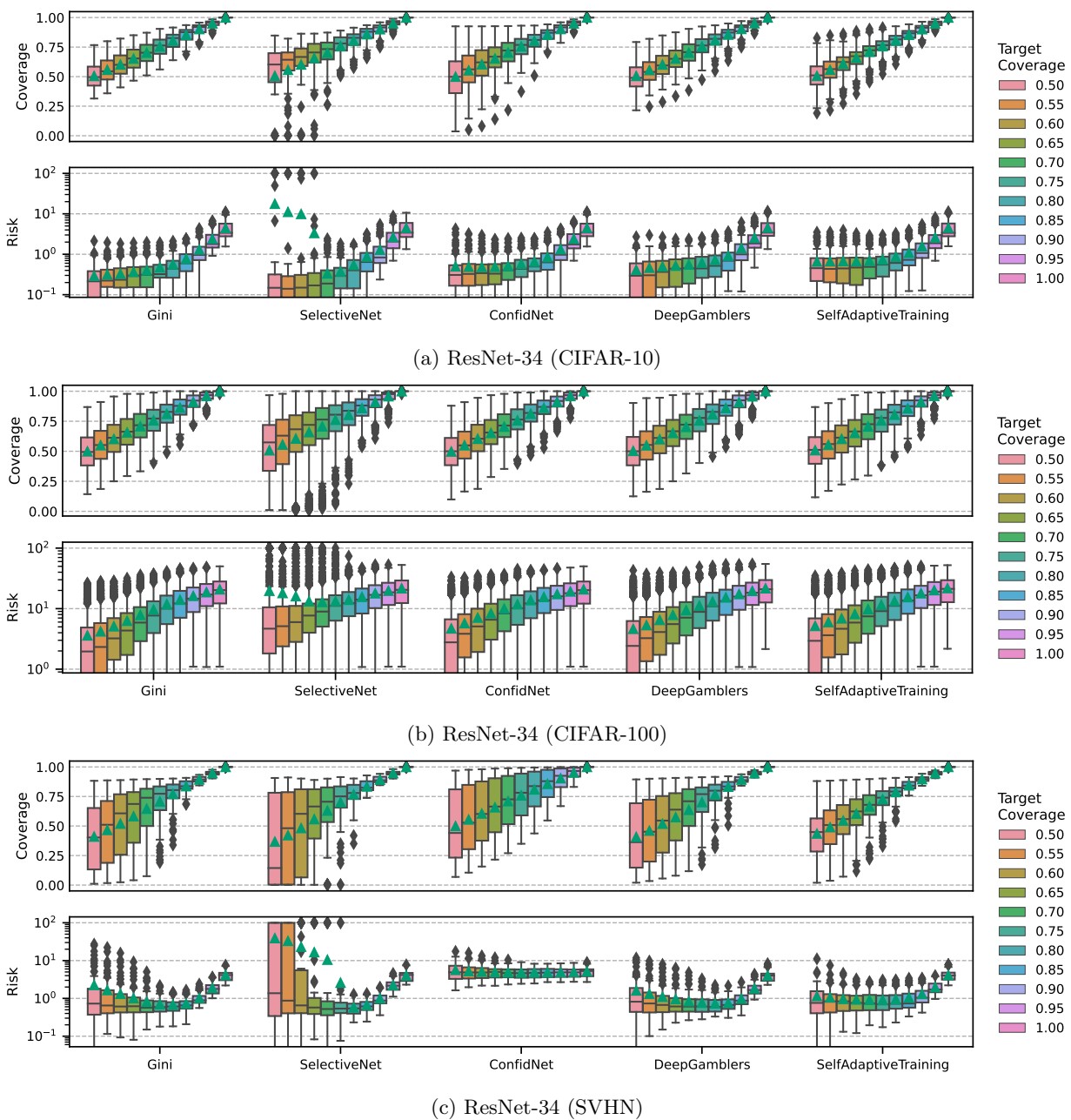

(a) ResNet-34 (CIFAR-10)

(b) ResNet-34 (CIFAR-100)

(c) ResNet-34 (SVHN)

Figure 9: Class-wise calibrated coverage and risk versus target coverage for a ResNet-34 model.

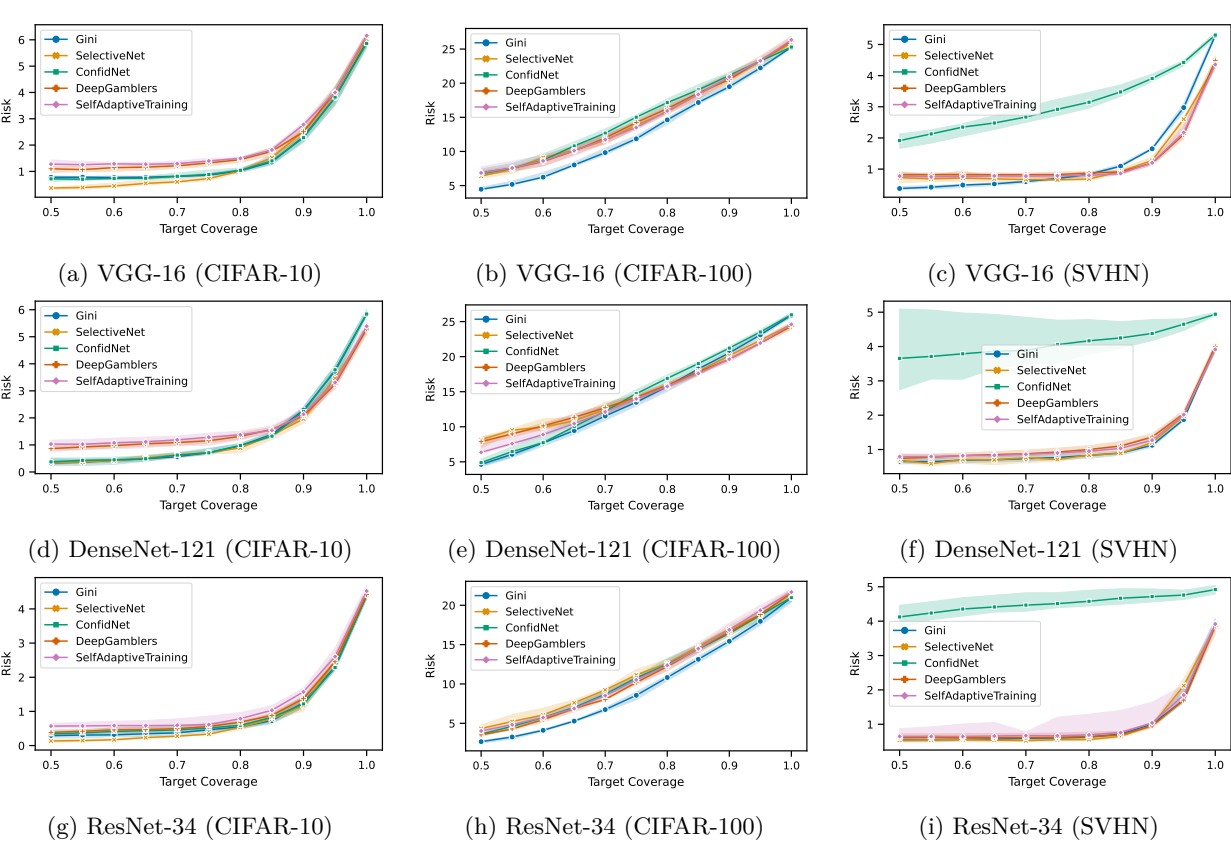

Figure 10: Global calibrated risk versus target coverage.

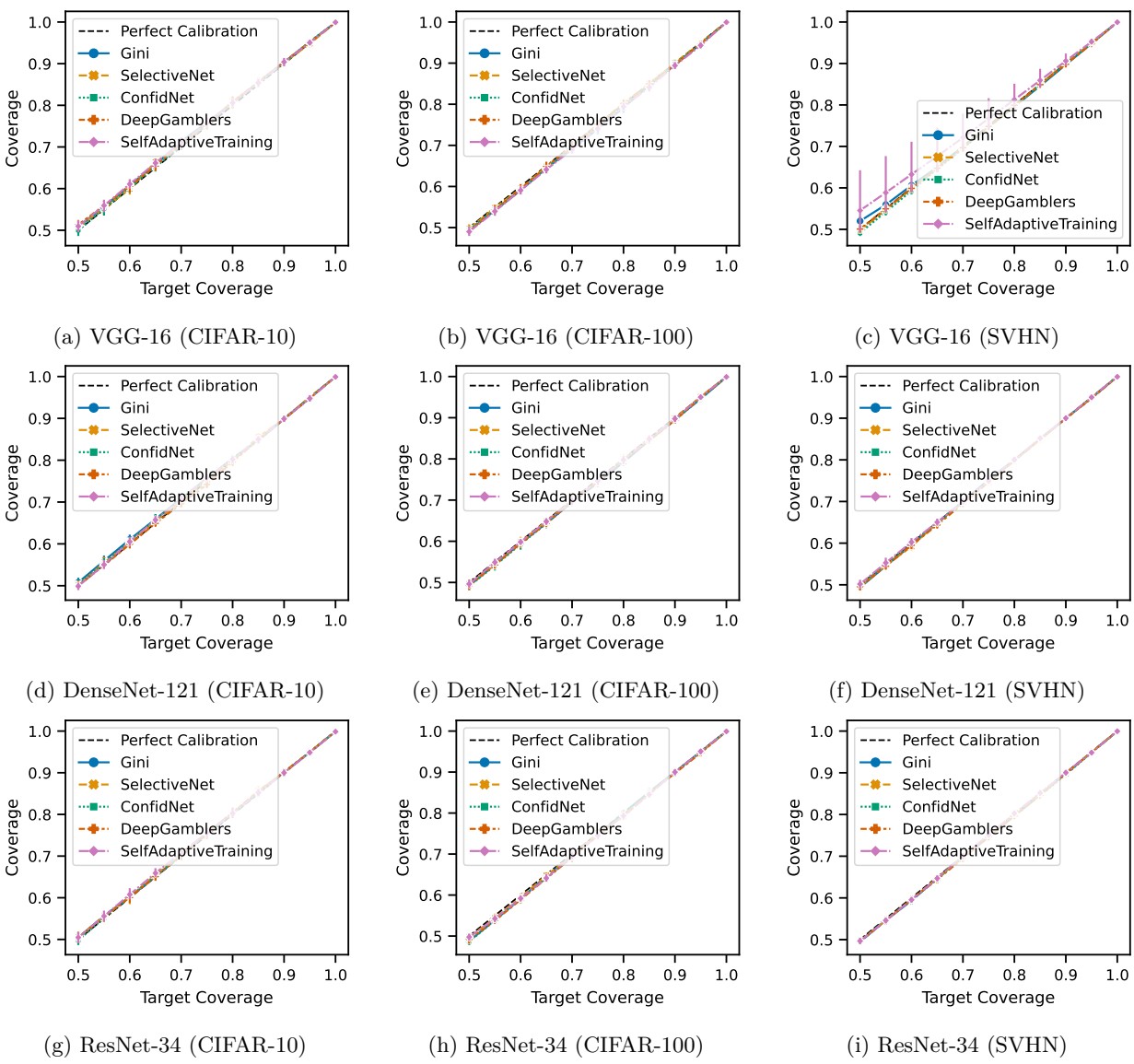

Figure 11: Global calibrated coverage versus target coverage.

