# OpenReview forum: "Trusting the Untrustworthy: A Cautionary Tale on the Pitfalls of Training-based Rejection Option"
_TMLR — Withdrawn by Authors_

### Review · Reviewer_Fvde · 2023-07-19

**Summary Of Contributions:**

The authors delve into the issue of classification problems with an option for rejection. This problem allows a classifier to utilize a rejection option to sidestep predictions that lack confidence. By doing so, the trustworthiness of the classifier can be enhanced as it may help to avert incorrect predictions. Then, the objective for the learner becomes achieving the optimal balance between the rate of rejection and the rate of misclassification within data that has not been rejected. To address this issue, the authors formulated a post-processing algorithm that constructs a function that determines whether an input should be rejected. Empirical evaluations have underscored the efficacy of their approach.

**Audience:**

No

**Broader Impact Concerns:**

I have no concern about a broader impact.

**Claims And Evidence:**

No

**Requested Changes:**

- I kindly suggest addressing the first weakness. It would be beneficial for the authors to include results that surpass the contributions made by Granese et al.
- Could you please review Proposition 4.2?
- It would be useful to update the results for the training-based methods using the version with tuned hyperparameters.
- A demonstration highlighting the advantages of the proposed method compared to ODIN would be highly appreciated.

**Strengths And Weaknesses:**

## Strengths

1. The reliability of machine learning predictions presents a pressing challenge that hinders the practical application of these techniques in real-world systems. The authors have developed a method to improve the trustworthiness of classification algorithms by effectively incorporating a 'reject' option. This approach should be of great interest to the audience of TMLR.

2. The proposed method involves constructing a rejection determination function post hoc, using a previously learned model that determines the posterior probabilities of labels. This approach can be seamlessly integrated with many classification algorithms that lack rejection options, highlighting its suitability for practical applications.

## Weaknesss

1. The primary concern with this manuscript is its significant overlap with the contributions made by Granese et al. in 2021. The proposed methodology mirrors that of Granese et al., and the absence of new findings may diminish its appeal to the TMLR audience.

2. The theoretical result of Proposition 4.2 may be incorrect due to errors in the derivations used in its proof. The authors implicitly state that the optimal minimum selection risk is achieved when $P_{I} + P_{II} = 1 - \|p_{X|E=1} - p_{X|E=0}\|\_{TV}$, equivalently when the selection classifier minimizes $P_{I} + P_{II}$. However, the authors have not provided any evidence to show that the minimizers of the selection risk and $P_{I} + P_{II}$ are the same. Consequently, the proof might not fully endorse the claims of Proposition 4.2.

3. The experimental comparisons appear to be unfair in the treatment of hyperparameters. The authors optimize their hyperparameters using the validation dataset specifically for Gini and ODIN. Conversely, they utilize a predetermined set of hyperparameters for the other training-based methods. This approach puts the training-based methods at a disadvantage, leading to an unfair comparison. The primary conclusion drawn by the authors from the experimental evaluations is the inferior performance of training-based methods compared to post hoc methods. This flawed comparison is crucial to the validity of this conclusion.

4. The experimental results do not show any significant advantage of the proposed method over ODIN. The selective risk of the proposed method falls within the confidence interval of ODIN's. The absence of a clear benefit when compared to ODIN might reduce the interest of the TMLR audience in the proposed method.

---

### Review · Reviewer_CCQM · 2023-07-23

**Summary Of Contributions:**

This paper introduces a more efficient selective classification method motivated by the limitations of previous approaches in achieving satisfactory performance across all classes. Although previous methods have demonstrated reasonable empirical risk and coverage of testing data, their performance remains low in certain classes. The paper aims to establish a mathematical relationship between P1 and P2 using the Gini index, which leads to the proposition of an "optimal misclassification detector." This detector achieves a favorable trade-off while minimizing the selective classification risk. They proposed a connection between this approach and previous research inspired by Granese et al. (2021). The paper compares its results with various existing methods such as SlecetiveNet, ConfidNet, DeepGambler, SelfAdaptiveTraining, MCDroup out, and ODIN, demonstrating promising performance.

**Audience:**

Yes

**Claims And Evidence:**

Yes

**Requested Changes:**

1-Provide Justification for Comparison Methods: The authors should offer a clear and detailed explanation of why they selected ODIN and MonteCarlo Dropout as the comparison methods. This should include a discussion of their advantages over other potential alternatives and why they are relevant choices for the evaluation.

2-Discuss Relationship with Open Set Recognition: The authors should acknowledge the importance of open set recognition and discuss how their proposed approach relates to or can be extended to handle scenarios involving unknown or uncertain samples during classification. This will show that their method is relevant and applicable to a broader range of real-world problems.

3-Address Assumptions and Limitations: The authors should openly acknowledge the assumptions and simplifications made in their method and discuss the potential limitations, such as the inability to handle OOD samples belonging to known classes. They should also explore ways to address these limitations, either through model modifications or additional techniques.

4-Conduct Robustness Experiments: To evaluate the practical applicability of their approach, the authors should perform comprehensive robustness experiments. These experiments should include testing the model's performance under various types of noise, distribution shifts, and perturbations commonly encountered in real-world scenarios.

5-Consider Practical Generalization: The authors should investigate how their method performs on OOD samples that still belong to one of the existing classes. They should explore ways to enhance the model's generalization capabilities, ensuring that it can accurately classify such samples despite being different from the training data.

6-Discuss Implications and Future Directions: In the discussion section of the paper, the authors should address the implications of their findings and potential future research directions. They should highlight the practical applications and real-world use cases of their proposed method, which will enhance the significance of their contribution.

**Strengths And Weaknesses:**

Strengths:

1-Addressing the trade-off between coverage, empirical risk, and misclassification error, which was largely overlooked in previous research. This shows a more holistic approach to the problem and highlights the importance of considering all these factors together.

2-Providing a clear and thorough explanation of previous works on selective classification and successfully establishing connections between this work and the earlier research. This contextualizes the new approach within the existing body of knowledge.

3-Introducing a  theoretical framework based on P1 and P2 phenomena and utilizing the Gini index. This theoretical support offers a fresh perspective on the problem, potentially paving the way for new insights and solutions.

4-Demonstrating superior reliability for all classes compared to previous methods that exhibited good overall performance but struggled in specific classes. This indicates that the proposed approach is more robust and capable of handling diverse data scenarios.

5-Conducting comprehensive experimental results, which further solidifies the credibility of the proposed method. The extensive evaluation of the approach against various existing methods helps validate its effectiveness and competitiveness.

Overall, the paper's strengths lie in its holistic approach to the problem, clear explanations, novel theoretical foundations, improved reliability across classes, and rigorous experimental validation. These strengths contribute to making it a valuable theoretical and practical contribution to the field.

Weaknesses:

1-The  lack of clear justification for selecting ODIN and MonteCarlo Dropout as the comparison methods. While the paper provides a comprehensive review of previous methods and compares them with these two techniques, it doesn't explain the specific reasons for choosing them over other potential alternatives. A more detailed rationale for the selection would enhance the validity of the comparative analysis.

2-The paper fails to establish a direct relationship with the growing area of research known as open-set recognition. Given the increasing importance of addressing the challenge of unknown or uncertain samples during classification, it would have been beneficial for the authors to discuss how their approach relates to or can be extended to handle open-set recognition scenarios.

3-Although the proposed method shows great theoretical performance and addresses an important problem, the paper makes certain assumptions or simplifications that may limit its practical applicability. For instance, the method might struggle when faced with out-of-distribution (OOD) samples that belong to one of the existing classes but differ significantly from the training data distribution. This limitation raises questions about the generalization capabilities of the model in real-world scenarios where noise or coverage shifts are prevalent.

4-The paper would benefit from including a more comprehensive analysis of the robustness of the proposed method to various perturbations and uncertainties in the input data. Evaluating the model's performance under different forms of noise or distribution shifts would provide a clearer understanding of its limitations and strengths in practical situations.

Addressing these weaknesses could lead to a more comprehensive and well-rounded research contribution. By providing clearer justifications for comparison methods, exploring connections to open set recognition, addressing limitations related to OOD samples, and conducting additional robustness experiments, the paper would strengthen its practical relevance and impact within the field.

---

### Review · Reviewer_MdA8 · 2023-08-15

**Summary Of Contributions:**

This paper tackles the problem of selective classification, i.e. classification for which the model is allowed to abstain from making a decision. Methods for selective classification are either post-hoc (in the sense that they can work with any pre-trained model) or training-based (in the sense that they require to fit a new model, with a new loss function fit for selective classification). The contributions of the authors are twofold:

1. They revisit several recent training-based methods and show empirically that they have several shortcomings, in particular that they may work well globally but can have poor performance for some classes. They investigate possible reasons for these issues by discussing the particulars of these methods

2. They propose a new post-hoc method that is quite simple (the selective score is just the Renyi entropy/Gini impurity) and competitive with state-of-the-art methods.

The main narrative used to derive the new method is as follows:
- the true (but unknown) probability of error $\text{Pe}(x) = \mathbb{P}(f_{\mathcal{D}_n}(x) \neq y)$ would be the optimal reject score because of Formula (13) in their Proposition 4.2
- thus, a good estimate of $\text{Pe}(x)$ would be a good reject score, and Granese et al. (2021) showed that using the Gini impurity/Renyi entropy is a good estimate of $\text{Pe}(x)$, therefore, the Gini impurity/Renyi entropy should be a good reject score.




**Audience:**

Yes

**Broader Impact Concerns:**

I have no particularly strong concerns here. However, I think that the problem of selective classification has several societal ramifications (e.g. in medicine, cf for instance Kompa et al., 2021) which are only discussed in a vague way in the paper. Being a bit more precise about these could be interesting.

Kompa et al., Second opinion needed: communicating uncertainty in medical machine learning, npj Digital Medicine, 2021

**Claims And Evidence:**

Yes

**Requested Changes:**

- Adding the standard maximum probability method as a baseline

- Clarifying my issues with the implementation of MC Dropout

- Clarifying the novelty of Proposition 4.2

- Discuss what can happen when there is no validation set



**Strengths And Weaknesses:**



**Strengths**

This paper is quite well-written, and reviews nicely prior work on the rejection option. I think that showing that a simple methods are competitive with more complex ones is an important message in general. The inspection of the characteristics of several methods (in Section 3) is quite nice, although a bit oversold at times (see the "minor comments" section). The experimental design is quite interesting, although I would have liked to see more datasets (e.g. smaller UCI datasets). The new Gini method appears to work well, and most methods have been included in a nice github repository (I have not run the code myself, but it looks reproducible).

**Weaknesses**

1) My main concern is related to an issue in the narrative I sketched in Section "Summary Of Contributions". Indeed, Granese et al. (2021) actually proposed *two* candidates to approximate  $\text{Pe}(x)$, one based on Gini (that they call $D_\alpha$), and one based on the even simpler maximum probability (that they call $D_\beta$). I think that trying out both scores instead of just the Gini-based would be very useful. Indeed, using the maximum probability score is the simplest baseline one could think of (and is related to the good old proposal of Chow, 1970). I think adding this simple baseline would make the paper more insightful.

2) I did not understand how the authors implement Monte Carlo dropout, and did not find the code for it in the companion GitHub repository. Monte Carlo dropout is usually used by averaging probabilities obtained for each stochastic forward pass of the network, leading to new, averaged probabilities. After averaging, how do you select which points to reject? You could use Gini together with Monte Carlo dropout, or even together with the maximum probability I advocated above, or even the regular entropy. The authors only cite the original paper by Gal and Ghaharamani, which says very little about classification with a rejection option. Gal was the last author of Filos et al. (2019) which was a paper specifically on the rejection option. Here they used the Shannon entropy with 100 MC samples. The authors of the paper under review use only 5 samples, which seems to be a very low number.


3) I have a hard time assessing the novelty an usefulness of Proposition 4.2. It seems that the main insight gained from it is that the optimal score for selective classification should be $\text{Pe}(x)$, but isn't it exactly the conclusion of Chow (1970) as well? The authors should clarify the differences between the two papers. For instance, they mention that "Chow (1970) only takes into account the oracle detector", but don't they also do this because their result involves the true $\text{Pe}(x)$  ? I think one difference is that their starting point is the selective risk, which is not the case of Chow? As a side note, I didn't find Chow's paper to be an easy read, but I really like Ripley's (1996) treatment of the same results (in particular in his second chapter). Since it's one of the few  standard ML textbooks that actually explore selective classification, it'd be nice to cite it and discuss how his results are related or not to Proposition 4.2.

4) All methods are used with the same calibration procedure to select the threshold. I think this is an excellent choice and makes things easier to compare. However, it would be nice to also discuss what can be done when there is no fresh validation set. Is there a natural threshold that will work decently even without tuning it (e.g. like the 50% threshold when using Chow's rule) ?


**Minor issues**

a. In Fig.1, the risk of Gini is high (and its coverage low) for the class "cat", so does it not have (to a lesser extent perhaps) the same issues as training-based methods

b. Page 4:  "the optimization (...) causes gradient descent to reach different local minima": isn't that the case for any deep learning method?

c. Page 5:  "the algorithm may assign different confidences to the same samples across iterations": isn't that the case for any deep learning method?

d. Page 6: "[the] classical cross-entropy loss [results in] models that converge to similar final performance despite being randomly initialized". This is partly true, and not really the case at the sample level (see e.g. Fort et al., 2019)

**References**

Ripley, Pattern Recognition and Neural Networks, CUP (1996)
Filos et al., A Systematic Comparison of Bayesian Deep Learning Robustness in Diabetic Retinopathy Tasks, arXiv:1912.10481 (2019)
Fort et al., Deep Ensembles: A Loss Landscape Perspective, NeurIPS (2019)

---

### Note · Authors · 2023-08-16

**Comment:**

We thank the reviewers for their feedback. The number of additional experiments requested is too high given the time allocated to provide answers, and in some cases, the requests are too vague to be addressed. Therefore we decided to withdraw the submission.

**Withdrawal Confirmation:**

I have read and agree with the venue's withdrawal policy on behalf of myself and my co-authors.